# Universal Properties of Activation Sparsity in Modern Large Language Models

**Filip Szatkowski**[1,2]     **Patryk Będkowski**[1]     **Alessio Devoto**[3]     **Jan Dubiński**[1,4]

**Pasquale Minervini**[5,6]     **Mikołaj Piórczyński**[1,2]     **Simone Scardapane**[3]     **Bartosz Wójcik**[7]

## Abstract

Activation sparsity is an intriguing property of deep neural networks that has been extensively studied in ReLU-based models, due to its advantages for efficiency, robustness, and interpretability. However, methods relying on exact zero activations do not directly apply to modern Large Language Models (LLMs), leading to fragmented, model-specific strategies for LLM activation sparsity and a gap in its general understanding. In this work, we introduce a general framework for evaluating sparsity robustness in contemporary LLMs and conduct a systematic investigation of this phenomenon in their feedforward (FFN) layers. Our results uncover universal properties of activation sparsity across diverse model families and scales. Importantly, we observe that the potential for effective activation sparsity grows with model size, highlighting its increasing relevance as models scale. Furthermore, we present the first study of activation sparsity in diffusion-based LLMs. Overall, our work provides a comprehensive perspective and practical guidance for harnessing activation sparsity in LLM design and acceleration.

## 1 Introduction

An intriguing property of deep learning models is activation sparsity, the tendency of their hidden states to contain a significant fraction of zero or near-zero values with input-dependent patterns. This phenomenon has been connected to robustness (Dhillon et al., 2018; Li et al., 2023), interpretability (Geva et al., 2021; Zhang et al., 2023; Cunningham et al., 2023; Bricken et al., 2023; Gao et al., 2024a), and potential efficiency gains from skipping redundant computations (Liu et al., 2023; Mirzadeh et al., 2024). Motivated by these benefits, prior work has studied activation sparsity extensively in ReLU-based networks (Glorot et al., 2011; Awasthi et al., 2024; Rhu et al., 2018; Kurtz et al., 2020; Zhang et al., 2021; Li et al., 2023).

However, the landscape of modern deep learning has shifted with the rise of Large Language Models (LLMs), which predominantly use GLU-based architectures (Shazeer, 2020) with SiLU or GELU activation functions (Gemma Team, 2024; Touvron et al., 2023; Qwen Team, 2023; DeepSeek-AI, 2024). Although the hidden states of these models often contain many near-zero values, methods developed for ReLU-based networks transfer poorly when applied to them directly. Recent works have attempted to adapt activation sparsity techniques to LLMs by retrofitting them to use ReLU or similar activation functions that induce exact zero activations (Zhang et al., 2021; Mirzadeh et al., 2024; Song et al., 2024a;b), or exploit approximate sparsity in existing architectures (Federici et al., 2024; Lee et al., 2024; Chua et al., 2024; Liu et al., 2025a;b). However, retrofitting approaches often require additional training and risk degrading model quality during such training, while approximate sparsity approaches lack the principled guarantees of strict sparsity induced by ReLU, require calibration of thresholds, and may overfit to the held-out calibration dataset. Moreover, the literature on sparsity-based acceleration remains fragmented, with different methods targeting various components of the LLM modules, such as feed-forward network (FFN) input (Federici et al., 2024; Liu et al., 2025a;b), gate (Song et al., 2024a; Lee et al., 2024), or intermediate activations (Liu et al., 2023; Akhauri et al., 2024). Consequently, the design choices behind LLM activation sparsity methods can seem arbitrary,

---

[1]Warsaw University of Technology, [2]IDEAS Research Institute, [3]Sapienza University of Rome, [4]NASK National Research Institute, Poland, [5]University of Edinburgh, [6]Miniml.AI, [7]Jagiellonian University, Kraków

Figure 1: Common strategies for exploiting activation sparsity to skip redundant computations in GLU-based FFN modules, with the origins of the sparse activation masks denoted with red borders. **Input-based** methods skip parts of the matrix multiplications corresponding to the low-magnitude components in the input vectors across all three linear layers. **Gate-based** and **predictor-based** methods instead omit computations associated with values that are either negligible in the gate activation vector or predicted to be negligible by an auxiliary predictor module.

and there is currently no comprehensive overview of the phenomenon in modern models, nor unified guidelines for leveraging it to enhance model efficiency.

Given the rapid advances in LLMs, we argue that a systematic study of activation sparsity in such models is necessary to understand their internal mechanisms, improve their efficiency, and guide their architectural design. In this work, we address this gap by analyzing the robustness of contemporary LLMs to activation sparsity across FFN components, architectures, and model sizes. We provide a unified view of sparsity patterns and introduce a simple, general framework for assessing model robustness. To induce sparsity, we propose a straightforward top-p method, applicable to any model without architectural assumptions or additional training. This allows us to assess models' activation sparsity from a functional perspective and identify the highest sparsity levels that do not trigger meaningful quality loss. We formalize this concept of functional activation sparsity via *critical sparsity*, a metric defined as the maximum sparsity level that causes at most a 1% performance drop. As prior work has already demonstrated that with appropriate implementation sparsity in FFNs activations can translate approximately linearly to computational acceleration (Szatkowski et al., 2024; Liu et al., 2025a), in our work we focus on characterizing models' tolerance to sparsity and uncovering universal patterns across popular LLMs.

Using our proposed framework we find that the critical sparsity tends to increase with model size, and present empirical comparisons of sparsity tolerance across FFN activations. We further examine the impact of training strategies on sparsity robustness, highlighting the universal capacity for activation sparsity in pretrained, instruction-tuned, and reasoning models. Additionally, we analyze activation sparsity and temporal consistencies in sparsity patterns in masked diffusion LLMs, revealing potential acceleration opportunities; to our knowledge, this is the first study of such phenomena in this class of models. Finally, we discuss our findings in the context of prior work and mention its implications for designing future methods that aim to leverage activation sparsity. Overall, our study uncovers universal patterns of activation sparsity in modern LLMs and offers practical insights and guidelines for researchers and practitioners seeking a deeper understanding of this intriguing phenomenon.

## 2 ACTIVATION SPARSITY IN MODERN TRANSFORMERS

### 2.1 SPARSE ACTIVATIONS IN DEEP NEURAL NETWORKS

Activation sparsity refers to the tendency of neural network hidden states to contain a substantial fraction of zero values, following input-dependent patterns. This phenomenon has been widely observed in ReLU-based architectures, including MLPs (Glorot et al., 2011; Davis & Arel, 2013) and CNNs (Rhu et al., 2018; Kurtz et al., 2020). More recently, Zhang et al. (2021) and Li et al. (2023) reported that standard training also induces significant activation sparsity in Transformer feed-forward networks (FFNs). Activation sparsity has been associated with several desirable properties. In MLPs, it can enhance learnability and generalization (Awasthi et al., 2024), while in CNNs and ViTs it has been shown to improve robustness against input corruptions (Ahmad & Scheinkman, 2019; Li et al., 2023). Moreover, sparsity has been leveraged for interpretability by disentangling neurons

corresponding to distinct concepts (Geva et al., 2021; Elhage et al., 2022; Cunningham et al., 2023; Bricken et al., 2023; Gao et al., 2024a).

Modern LLMs typically use SiLU or GELU activations in combination with GLU-based FFNs (Shazeer, 2020), and do not exhibit strictly zero activations. Nevertheless, effective activation sparsity still arises in these models even without explicit architectural or regularization constraints, and several studies have sought to theoretically explain this phenomenon (Andriushchenko et al., 2023; Peng et al., 2023). Zhang et al. (2024b) investigated how different activation functions during LLM pretraining influence emergent sparsity patterns, while Luo et al. (2024) derived scaling laws connecting activation sparsity to the size of pretraining datasets. Additionally, some works have explored leveraging the approximate sparsity in LLM activations to improve model efficiency (Liu et al., 2023; Mirzadeh et al., 2024), which we discuss in more detail in Section 2.3. Despite these efforts, most studies focus on particular forms of activation sparsity, and a unified, systematic investigation of the phenomenon in modern LLMs is still lacking.

## 2.2 ACTIVATION VECTORS IN GATED FEEDFORWARD TRANSFORMER LAYERS

Transformer blocks consist of attention and feedforward (FFN) sub-blocks (Vaswani et al., 2017). While the original Transformer FFN was composed of two projection layers separated by an activation function, modern LLMs typically employ FFNs based on the Gated Linear Unit (GLU) architecture (Shazeer, 2020), which can be expressed as:

$$\mathcal{FFN}(x) = \mathbf{W_d}\big((\mathbf{W_u}x) \odot \sigma(\mathbf{W_g}x)\big),$$

where $x \in \mathbb{R}^h$ is the input vector, $\mathbf{W_u} \in \mathbb{R}^{h \times d}$ is the up-projection matrix, $\mathbf{W_g} \in \mathbb{R}^{h \times d}$ is the gating projection matrix, $\mathbf{W_d} \in \mathbb{R}^{d \times h}$ is the down-projection matrix, and $\sigma$ is an activation function, usually SiLU or GELU. We use $h$ and $d$ to denote the model's hidden and intermediate dimensions, respectively. In the subsequent sections, we refer to the above-mentioned activation vectors in the FFN as $x$ - *input*, $u = \mathbf{W_u}x$ - *up-projection*, $g = \sigma(\mathbf{W_g}x)$ - *gate* and $i = (\mathbf{W_u}x) \odot \sigma(\mathbf{W_g}x)$ - *intermediate* vectors. Additionally, we consider the simultaneous sparsification of input and intermediate activations, and refer to the case where we induce sparsity in both of these vectors at the same time as *all FFN inputs* sparsification.

## 2.3 LLM INFERENCE ACCELERATION WITH ACTIVATION SPARSITY

Modern models exhibit significant activation sparsity, meaning that a large portion of activations are effectively unused, resulting in wasted computation. This effect is especially pronounced in FFN layers, which dominate the computational cost of LLMs until extremely long contexts (Casson, 2023). Skipping these redundant computations can not only reduce computation, but also lower memory overhead, decrease the cost of loading weights, and even enable hardware-specific optimizations such as weight caching (Alizadeh et al., 2024; Federici et al., 2024). LLM acceleration methods that leverage activation sparsity can be grouped into three main categories outlined in Figure 1.

**Input-based methods** (Federici et al., 2024; Liu et al., 2025a;b) select a subset of input activations for each linear layer and skip the operations corresponding to masked out multiplications. Since input vectors might lack natural sparsity, e.g., from activation functions, these methods calibrate skip thresholds or use heuristics to identify redundant activations. **Gate-based methods** (Lee et al., 2024; Song et al., 2024a) exploit sparsity based on the gate activation vector. Since this vector is a result of projection through an activation function, such methods assume that it should contain many near-zero values, which render portions of the up- and down-projection computations redundant. However, computing the gate vector accounts for roughly one-third of the FFN cost, which limits its overall efficiency. **Predictor-based methods** (Zhang et al., 2021; Liu et al., 2023; Szatkowski et al., 2024) usually target the intermediate activations. Computing intermediate activation vectors directly requires over two-thirds of FFN computation, so these methods typically use a lightweight predictor (e.g., a small linear network) instead to estimate which neurons (or groups) can be skipped. This approach can reduce computations in all FFN components, but requires training the predictor, induces slight inference overhead on the prediction, and introduces potential approximation errors.

Research on activation sparsity typically targets specific types of activations and focus on improving model efficiency. Prior studies have already demonstrated computational benefits of activation

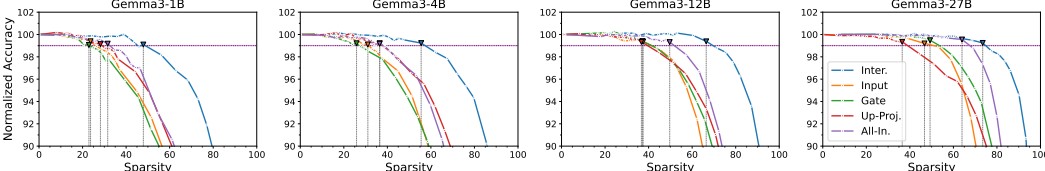

(a) Normalized accuracy and activation sparsity of FFN modules in Gemma3 models across the model sizes.

(b) Normalized accuracy and activation sparsity for different sizes of Gemma3 across the FFN modules.

Figure 2: Average accuracy across downstream tasks with different induced activation sparsity for base Gemma3 models. We normalize the accuracy by the original performance of the dense models, and denote the highest (*critical*) sparsity where at least 99% performance is retained with a marker.

sparsity for dense FFN computation: with a sufficiently optimized kernel that can exploit sparsity in the linear computations, both FLOPs and execution latency of the linear modules scale approximately linearly with activation sparsity (e.g., see Figure 3 in Liu et al. (2025a) and Figure 5 in Szatkowski et al. (2024)). Since these computational gains are already well-established, our work instead focuses on developing a broader understanding of activation sparsity across modules and model architectures.

## 3   DETERMINING ACTIVATIONS TO SPARSIFY IN NON-RELU LLMS

Modern LLM architectures lack components that explicitly produce zero activations, which makes it difficult to study activation sparsity directly. However, prior work (Chua et al., 2024; Liu et al., 2025a;b) has shown that some activation vectors $v \in \mathbb{R}^n$ in such models can be *sparsified* to a certain degree without incurring a significant performance loss. To study the general impact of sparsification on FFN layers, we propose to use a simple top-p sparsification rule, where we obtain a sparsity mask $m_p$ from the largest-magnitude entries in $v$ whose absolute values sum to at least a fraction $p$ of the vector's total L1 norm:

$$\text{top-p}(v) = m_p \odot v; \ m_p = \arg\min_m ||m||_0 \ \text{s.t.} \ ||m \odot v||_1 \geq p \cdot ||v||_1 \text{ and } m \in \{0,1\}^n.$$

The induced sparsity is then the fraction of zeros in $m_p$: $S_p(v) = \frac{1}{n} \sum_{i=1}^n \mathbb{1}(m_p^{(i)} = 0)$. By evaluating model performance over a range of $p$ values, we can obtain a sparsity-performance trade-off curve and assess the *functional activation sparsity* of the model – the level of sparsity at which the model still performs comparably well to the densely activated original. To this end, we introduce the concept of *critical sparsity* - the highest sparsity level at which the model retains at least 99% of the performance. This notion provides us with a practical way to characterize a model's ability to tolerate activation sparsity while anchoring the analysis in realistic performance constraints.

Our approach is simple, general, and easy to interpret. Crucially, it can be applied to any FFN module without requiring auxiliary training or calibration, which enables a fair comparison of models and modules. See Appendix A for further discussion and comparison between top-p and the alternatives.

## 4   EXPERIMENTS

To study the effects of activation sparsity in LLMs, we evaluate Gemma3 (Gemma Team, 2025), Llama3.1/3.2 (Meta AI, 2024), and Qwen2.5 (Qwen Team, 2024) models using lm-eval-harness (Gao et al., 2024b) in a zero-shot setting. Unless otherwise specified, we use pretrained model variants and report the average performance across all the tasks from the task suite

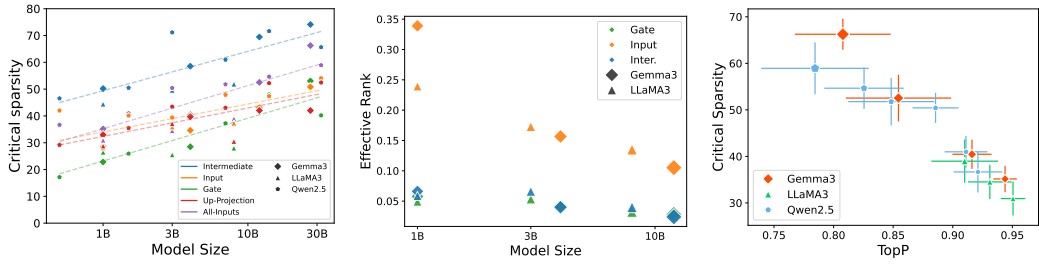

(a) Model size and critical sparsity.  (b) Activation effective ranks.  (c) Sparsity and corresponding $p$.

Figure 3: Activation sparsity becomes more pronounced as model size increases. a) Average critical sparsity of FFN components across models, with least-squares trend lines. Larger models generally tolerate higher sparsity, suggesting greater potential benefits. b) Effective ranks (Roy & Vetterli (2007)) of activations on Winogrande, normalized by activation dimension. Larger models show lower effective dimensions, indicating greater redundancy available for sparsification. c) Critical sparsity under All-Inputs sparsification and the corresponding top-p thresholds at which performance degrades. Results are averaged across evaluation tasks, with marker size indicating model size.

from Mirzadeh et al. (2024), which includes: ARC-Easy (Clark et al., 2018), ARC Challenge (Clark et al., 2018), PIQA (Bisk et al., 2020), BoolQ (Clark et al., 2019), HellaSwag (Zellers et al., 2019), WinoGrande (Sakaguchi et al., 2021), Lambada (Paperno et al., 2016), SciQ (Welbl et al., 2017) and TriviaQA (Joshi et al., 2017) datasets. For clarity, throughout our study, we primarily focus on Gemma models, as they exhibit the most uniform dimension and depth scaling with model sizes. We provide the corresponding results for LLaMa and Qwen in the appendices.

To measure the relation between sparsity and model performance for a given activation type from Section 2.2 (input, gate, up-projection, intermediate, or all inputs), we apply the top-p rule uniformly across all layers over a range of thresholds $p$. For each threshold, we then measure the average induced sparsity and the corresponding performance drop, thereby obtaining the empirical relationship. To explicitly link sparsity with accuracy, we focus primarily on the *critical sparsity* introduced in Section 3 - the highest empirical sparsity level at which models retain at least 99% accuracy. To compare the performance drop induced by activation sparsity applied to different models and tasks, we plot their accuracies normalized by the original performance.

## 4.1 ACTIVATION SPARSITY ROBUSTNESS IN FFN COMPONENTS

We begin our analysis by investigating the sparsification robustness of different types of activations in Gemma3 models. Specifically, we assess performance degradation under increasing sparsity induced by the top-p rule and show the results in Figure 2. Among the FFN components, intermediate activations demonstrate the highest tolerance to sparsity. However, as discussed in Section 2.3, leveraging this sparsity for computational acceleration in practice requires a sparsity predictor, which limits its practical benefits. Nevertheless, our results confirm that with sufficiently accurate predictors, exploiting intermediate sparsity remains the most promising strategy for reducing computation.

Interestingly, we observe that simple input-based methods, which use either the FFN input alone or combined with the down-projection input, achieve sparsity levels comparable to or exceeding those of the gate-based approach often favored in prior work (Song et al., 2024a; Lee et al., 2024). Although the gate activation naturally shrinks activations, its sparsity does not surpass input sparsity; in practice, raw input and up-projection sparsity behave similarly to gates. This makes input-based sparsification the most practical predictor-free approach, as it can accelerate all FFN modules without introducing approximation error. By contrast, gate-based sparsification offers no clear advantage at the model scales studied here, though it may surpass input-based methods in models larger than ~30B parameters. Overall, we find that the sparsity robustness of all FFN components generally improves with model size, a trend we explore in more detail in the following section.

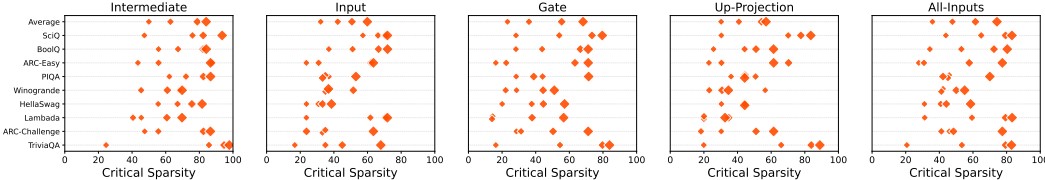

Figure 4: Critical sparsity for Gemma3 models (1B, 4B, 12B, and 27B) across all evaluated modules and tasks. Marker size represents model scale, and tasks with higher accuracy are positioned toward the top. While tasks with higher baseline accuracy generally tolerate sparser activations, critical sparsity varies substantially across tasks, highlighting that activation sparsity is task-dependent.

## 4.2 MODEL SIZE AND ACTIVATION SPARSITY

**Critical sparsity against model size.** To assess the generality of our previous findings across families and scales, we consider pretrained LLaMA3.1/3.2 and Qwen2.5 models and plot their critical sparsity in Figure 3a, fitting trends with least squares (see Appendix B for the table with the exact values). The trends outlined in the previous section remain roughly consistent across model sizes: intermediate activations are generally the most sparse, and the gate sparsity is comparable with input or up-projection sparsity until the larger model sizes. We attribute the slight deviations in the trends to the non-uniform depth–width scaling, particularly prominent in the Qwen model family, where dimensions grow disproportionately with parameter count. Overall, activation sparsity tends to increase with model size, though it cannot be directly determined based on the model size alone.

**Effective ranks of the model activations.** We further examine activation sparsity through effective ranks (Roy & Vetterli, 2007) of activations in Gemma and LLaMA models on the Winograd task. In this experiment, we do not sparsify activations, and instead compute their effective rank across layers and report the mean value. In Figure 3b, we show the effective ranks of intermediate, input, and gate activations. To enable comparisons between different modules and models, we normalize the ranks by the activation dimensionality (see Appendix E for computational details). Effective ranks consistently decrease with model size, reinforcing the observation that larger models exhibit greater capacity for activation sparsity. However, interestingly, gate activations exhibit effective ranks comparable to intermediates, despite their lower empirical sparsification capacity. This suggests that effective rank alone may be insufficient to fully capture robustness to sparsification.

**Relationship between critical sparsity threshold $p$ and model size.** Finally, we investigate how the critical sparsity threshold $p$ varies with model size. In Figure 3c, we plot the relationship between the critical sparsity and $p$ for all-inputs sparsification. Both the threshold and critical sparsity are averaged across all evaluation tasks, with marker size indicating model size. While the critical sparsity for a given model generally varies across tasks, larger models consistently tolerate lower $\text{top-p}$ thresholds and exhibit higher critical sparsity. Therefore, as model size increases, maintaining performance requires a smaller fraction of the total activation norm and a smaller fraction of the overall activations. Together with the previously observed scaling of effective rank, these results have significant implications for efficiency and sparsification, indicating that larger models have an increasingly pronounced long tail of neurons that do not significantly contribute to their performance.

## 4.3 DOWNSTREAM TASK IMPACT ON ACTIVATION SPARSITY PATTERNS

Recent works on activation sparsity for LLM acceleration (Liu et al., 2023; Lee et al., 2024; Liu et al., 2025a;b) typically assume that sparsity predictors or activation thresholds, tuned on a small dataset, generalize reliably to downstream tasks. However, activation sparsity patterns are by definition dynamic and input-dependent, which calls this assumption into question. To study how sparsity robustness varies across tasks, we analyze the critical sparsity of the FFN modules in four Gemma variants (1B, 4B, 12B, and 27B parameters, indicated by marker size) in Figure 4. We report performance on separate evaluation tasks, as well as the average across tasks, and order the tasks by the dense model's accuracy, with higher-accuracy ("easier") tasks at the top.

---

Comparable results for LLaMA and Qwen models are shown in Appendix D.

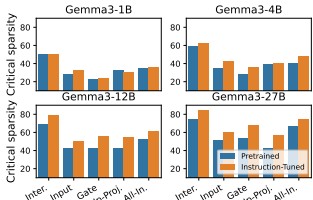 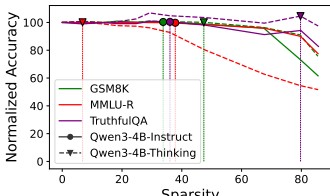 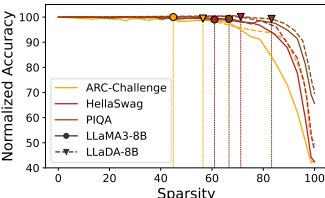

(a) PT and IT models comparison.     (b) IT and Thinking Qwen3-4B.     (c) Sparsity in LLaDA vs LLaMA.

Figure 5: Activation sparsity is a prevalent property across different model types. a) Critical sparsity levels for pretrained and instruction-tuned Gemma3 models. b) Performance of instruction-tuned and reasoning variants of Qwen3-4B on generative tasks assessing general knowledge, mathematics, and factual accuracy, with critical sparsity indicated by the markers. c) Activation sparsity in LLaMA-8B and the diffusion-based LLaDA-8B, with critical sparsity similarly marked. We report normalized accuracy as the accuracy of sparsified model divided by the accuracy of the dense version.

While the tasks that exhibit higher accuracy tend to be more robust to sparsification, there is no clear trend, and a model's critical sparsity generally varies per task. This suggests that downstream applications may be differently sensitive to sparsification, challenging the assumption that sparsification rules calibrated on a held-out dataset will generalize universally.

## 4.4 MODEL TRAINING AND ROBUSTNESS TO ACTIVATION SPARSITY

As established in the previous section, the model's activation sparsity depends on its architecture, size, sparsification method, and downstream task. The relation between the critical sparsity and the training recipe, however, remains underexplored. To address this, we study activation sparsity in instruction-tuned and reasoning variants of the models.

**Critical sparsity in pretrained and instruction-tuned models.**   We analyze average critical sparsity achieved across our task suite for pretrained Gemma3 models with that of instruction-tuned variants in Figure 5a. At larger scales, instruction-tuned models exhibit greater tolerance to activation sparsity, indicating that the training recipe can substantially influence robustness even when the underlying architecture is unchanged. The differences between pretrained and instruction-tuned models are consistent across all tested architectures, with detailed results provided in Appendix B.

**Reasoning with Qwen-4B.**   We explore activation sparsity in reasoning models, focusing on Qwen3-4B, a lightweight and cost-efficient model available in both Instruct and Thinking variants that can be fairly compared. We evaluate the performance of these models on three representative generative tasks: MMLU-Redux (knowledge) (Gema et al., 2024), GSM8K (math) (Cobbe et al., 2021), and TruthfulQA (factuality) (Lin et al., 2021). We report normalized accuracy on the tasks as a function of activation sparsity in Figure 5b. The reasoning model shows slightly greater robustness on GSM8K and even modest improvements with moderate sparsity on TruthfulQA, suggesting potential benefits of activation sparsity for the robustness. However, the performance of the reasoning model on MMLU-Redux degrades more sharply, as higher sparsity tends to lengthen outputs, causing them to exceed the maximum generation limit imposed by our compute constraints.

Our results demonstrate that activation sparsity consistently emerges across multiple post-trained models, independent of their training recipe. In light of these findings, activation sparsity appears to be a particularly promising yet relatively underexplored research avenue for enhancing the robustness and efficiency of reasoning models, which are becoming increasingly popular in modern applications.

## 4.5 EXPERT-WISE SPARSITY IN MIXTURE-OF-EXPERT MODELS

Sparse Mixture-of-Experts (MoE) models (Shazeer et al., 2016) have become a common choice for large-scale LLMs due to their efficient capacity scaling, and many recent frontier systems adopt this design (DeepSeek-AI, 2024). In these architectures, a sparse MoE layer replaces the standard FFN block with multiple experts, only a small subset of which is selected for any given input. Although

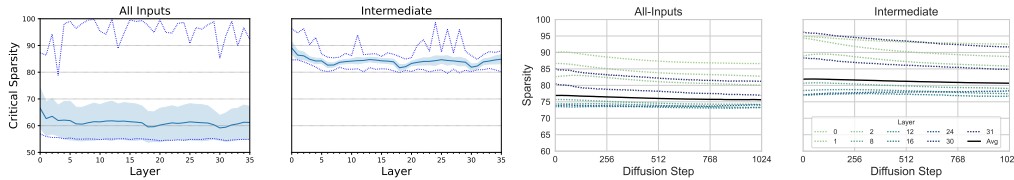

(a) Critical sparsity across Qwen3-30B-A3B layers.    (b) Activation sparsity across the diffusion steps.

Figure 6: a) Critical sparsity in Qwen3-30B-A3B Mixture-of-Experts model layers. We show the critical sparsity for each layer averaged over experts, alongside the minimum and maximum values across 128 experts in each layer. b) Critical sparsity of selected LLaDA-8B layers under All-Inputs and Intermediate sparsification on HumanEval. Overall activation sparsity remains relatively stable across diffusion steps. Interestingly, both early and late layers demonstrate the highest potential for sparsification, whereas middle layers exhibit slightly lower critical sparsity.

sparsity is usually discussed in terms of expert routing, each expert is still a standalone FFN and exhibits its own activation sparsity. This naturally leads to the question of how activation sparsity varies across experts within the same layer, and how pronounced these differences are in practice.

We study these questions using the Qwen3-30B-A3B model, a 30B-parameter MoE with 3B parameters active per forward pass (8 out of 128 experts per token). Using our standard evaluation suite, we measure the average critical sparsity across experts in each layer, along with the per-layer minimum and maximum sparsity of any single expert and the corresponding variance. We show the results for All-Inputs and Intermediate sparsification in Figure 6a. While layer-wise average sparsity across the experts stays relatively stable, individual experts within each layer exhibit notably higher, outlier sparsity levels. Interestingly, these critical sparsity levels exceed dense models and show surprising stability in the intertask variance, as shown in Appendix F. Overall, the results show that MoE experts reach activation sparsity levels comparable to dense LLMs, further reinforcing the universal existence of activation sparsity in modern LLMs and proving its viability for improving MoE's efficiency.

### 4.6 Activation Sparsity in Diffusion LLMs

Finally, we investigate activation sparsity in the increasingly popular class of diffusion-based LLMs. While prior work has explored sparsity and caching in image diffusion models (Ma et al., 2024; Silveria et al., 2025; Zhang et al., 2024a; Li et al., 2024a), to our knowledge this is the first analysis of activation sparsity in diffusion-based language models. Masked diffusion LLMs generate full sequences in parallel by progressively denoising across multiple diffusion steps, with each step producing distinct intermediate activations. To apply our sparsification scheme to such models, we compute separate sparsity masks at every model forward pass, and report sparsity averaged across all forward passes. We use the official implementation of recently introduced LLaDA-8B (Nie et al., 2025), which adopts the same architecture as autoregressive LLaMA3-8B. This choice enables fair comparisons without additional interference that would stem from the differences in model design.

Table 1: Critical sparsity values for diffusion-based LLaDA-8B.

|  | Intermediate | All-Inputs |
|---|---|---|
| **General Tasks** | | |
| MMLU | 69.46 | 62.72 |
| ARC-C | 56.48 | 31.74 |
| HellaSwag | 71.21 | 67.92 |
| WinoGrande | 50.75 | 42.35 |
| PIQA | 83.25 | 75.68 |
| **Maths and Science** | | |
| GPQA | 62.50 | 45.94 |
| **Coding** | | |
| HumanEval | 81.25 | 77.89 |
| MBPP | 66.67 | 59.18 |
| **Chinese** | | |
| CMMLU | 69.01 | 57.25 |
| C-Eval | 69.25 | 50.62 |
| Average | 68.13 | 56.79 |

**Critical sparsity in diffusion LLMs.** To compare activation sparsity in the autoregressive and diffusion LLMs, we investigate intermediate activations in LLaMA3.1-8B and LLaDA-8B. We examine sparsity characteristics across three tasks from our previous experiments: ARC-Challenge, HellaSwag, and PIQA, which are also analyzed in the LLaDA paper, and show the results in Figure 5c. Similar to the autoregressive model, LLaDA exhibits substantial activation sparsity and shows even slightly more favorable sparsity–performance trade-offs. To investigate the phenomenon in more detail, we measure critical sparsity (averaged over diffusion steps for generation or Monte-Carlo likelihood trials for scoring tasks) on tasks from the LLaDA evaluation suite, with both intermediate-and all-inputs activation sparsification. We present the results in Table 1. Average critical sparsity

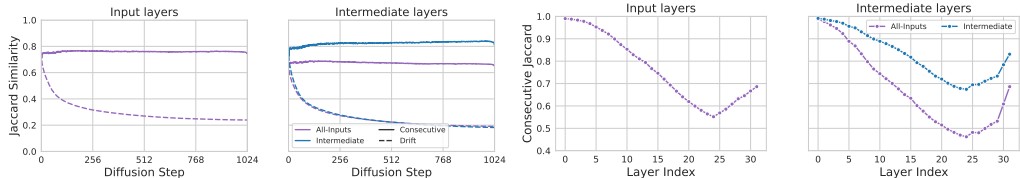

(a) Jaccard similarities in each diffusion step.      (b) Consecutive Jaccard similarity per model layer.

Figure 7: Jaccard similarity of the sparsity masks induced with critical sparsity in LLaDA-8B on HumanEval. We sparsify intermediate and both intermediate and FFN input activations, and plot the metrics for input and intermediate layers separately. a) Mask similarity between consecutive diffusion steps and between the current and the first step. We average the similarity across all modules of a given type. b) Consecutive similarity of each model layer, averaged over the whole diffusion process.

across all tasks is substantially higher than that observed in autoregressive LLaMA models in Figure 3a, likely due to the denoising nature of diffusion-based generation, which can better tolerate noise introduced by sparsification. Although activation sparsity remains largely unexplored in diffusion LLMs, our results indicate its considerable potential to accelerate computation in such models, to an even greater extent than in the autoregressive models.

**Sparse activation patterns across diffusion steps.** Finally, we analyze the temporal dynamics of activation sparsity in LLaDA. Previous work on autoregressive models has highlighted that sparsity patterns remain relatively stable after the prompt (Dong et al., 2024; Federici et al., 2024), which can be leveraged to make the model inference more efficient, for instance by reducing GPU weight-loading operations. We investigate whether similar efficiency techniques could be applied to diffusion-based LLMs by analyzing sparsity masks across diffusion steps to assess the presence of comparable temporal stability.

First, we look at the overall activation sparsity levels in LLaDA-8B across the diffusion steps on HumanEval dataset. For each diffusion step $t$, we apply the top-p rule introduced in Section 3 with fixed $p$ corresponding to the critical sparsity values obtained earlier, and obtain sparse masks $m_{p,t}$ for each diffusion step. We perform this experiment for both intermediate and all-inputs sparsification, and report the average sparsity level for each diffusion step across the whole dataset in Figure 6b. While the average activation sparsity slightly decreases throughout the denoising process, every step remains highly sparse at around 80% critical sparsity on average across all layers. Interestingly, we observe that both the earliest and the latest layers show the highest critical sparsity, while the middle layers exhibit slightly less sparsity.

The average activation sparsity remains high throughout the steps of the diffusion process, but that alone does not capture how stable the underlying activation patterns are. To measure this, we compute the Jaccard similarity between the sets of neurons active for two time steps $t_1$ and $t_2$ as:

$$J_{t_1,t_2} = \frac{|m_{p,t_1} \cap m_{p,t_2}|}{|m_{p,t_1} \cup m_{p,t_2}|}.$$

We measure both *consecutive Jaccard similarity*, where $t_2 = t_1 - 1$, and *drift Jaccard similarity*, where we fix $t_2 = 0$. Consecutive Jaccard measures local stability between adjacent diffusion steps, while drift Jaccard measures how sparsity patterns deviate from the initial sparsity mask.

In Figure 7a, we report the consecutive and drift Jaccard similarities computed over input and intermediate activation vectors, averaged across all model layers. While consecutive Jaccard similarity remains stable, drift similarity rapidly declines across diffusion steps, indicating that sparsity patterns undergo gradual yet substantial changes as the denoising process progresses. Notably, when sparsification is applied only to intermediate activations, the similarity metrics are higher, highlighting that sparsifying both input and intermediate activations makes the sparsity patterns less stable. In Figure 7b, we further present the average consecutive similarity of sparse activation sets across all diffusion steps, analyzed per LLaDA layer. Interestingly, consecutive similarity decreases steadily until approximately four-fifths of the model depth, after which it slightly increases near the final layer. Overall, despite the diffusion model maintaining consistently high overall sparsity across the

denoising process, the similarity between consecutive masks remains too low to allow for their reuse across steps, with potential exception of very early layers which exhibit very high Jaccard similarity.

**Sparsification in diffusion LLMs.**    To the best of our knowledge, our work is the first to demonstrate that activation sparsity is a highly prevalent phenomenon in diffusion-based LLMs. Our results suggest that sparsity could be a promising avenue for accelerating such models, as the critical sparsity levels in LLaDA significantly exceed the sparsity achieved with autoregressive LLaMA3.1 of the same size. However, our analysis of temporal sparsity patterns indicates that the behaviors observed in diffusion LLMs do not always mirror those seen in autoregressive models. Therefore, effective sparsification techniques for diffusion LLMs are likely to require model-specific solutions that leverage the unique properties of this new class of models.

## 5    CONCLUSIONS AND DISCUSSION

Despite lacking any architectural bias toward explicitly sparse activations, modern LLMs consistently exhibit functional sparsity. **We argue that functional sparsity is a universal property of LLMs and advocate for its wider adaptation when designing efficient models.**

We find that larger models tend to exhibit higher sparsity, suggesting that frontier models will become sparser as scaling continues. Therefore, **activation sparsity stands out as a promising tool for accelerating ever-growing LLMs**, and it already starts to appear in common model families, such as the recent Gemma3n release that includes activation sparsity-aware layers (You et al., 2025).

Our results show that input activations match or exceed the sparsity of gates and up-projections. Computing gates to choose sparsity patterns (Lee et al., 2024) is wasteful if they are no sparser than inputs, and newer work (Liu et al., 2025a; Federici et al., 2024; Liu et al., 2025b) demonstrates stronger acceleration with purely input sparsity. Overall, **our results suggest that input sparsification is the most practical training-free approach to leveraging activation sparsity for model acceleration**.

The high variance of critical sparsity across evaluation tasks and training recipes calls into question methods that rely on extra training (Zhang et al., 2021; Liu et al., 2023; Szatkowski et al., 2024) or threshold calibration (Lee et al., 2024; Liu et al., 2025a) on auxiliary datasets. Our results suggest that **sparsification methods should be truly data-free, as both functional sparsity levels and resulting patterns can be prone to overfitting**.

Our results should be seen as a lower bound on activation sparsity, as we adopt a simple, broadly applicable framework. While layer- or module-specific methods may achieve higher sparsity, our top-$p$ approach already reaches practical levels comparable to existing work. Given this and our earlier arguments on overfitting, **we argue that sparsification method design should favor simplicity**.

Our work is also the first to examine functional sparsity in diffusion LLMs. We highlight sparsity as a promising avenue for improving their efficiency, and expect that **activation sparsity could see increasing adoption in diffusion LLMs as their development advances**.

We focus on activation sparsity in feed-forward network (FFN) layers, excluding multi-head attention (MHA) to isolate the effect. In large models ($\geq$8B parameters), FFNs dominate inference costs for typical context lengths, and the cost of MHA only becomes significant for extremely long sequences (Casson, 2023). Even in such cases, techniques like KV-cache compression more effectively accelerate MHA due to its quadratic cost, making activation sparsity for attention less practical.

Finally, **we argue that evaluations of activation sparsity methods should prioritize performance preservation**, as captured by our notion of *critical sparsity*. Effective speedups from activation sparsity are limited to roughly 1.3–1.5$\times$ (Lee et al., 2024; Liu et al., 2025a;b), while techniques such as speculative decoding enable lossless speedups of up to 4$\times$ (Li et al., 2024b; 2025). Consequently, activation sparsity should be viewed as complementary to other acceleration methods rather than as the sole focus of efficiency studies on model acceleration techniques.

We have systematically analyzed activation sparsity in LLMs, demonstrating that functional sparsity is pervasive and tends to increase with model scale. We hope our work highlights the potential of activation sparsity for efficiency and provides insights for future model design. Our findings emphasize the growing significance of activation sparsity, and we expect that, as models continue to scale, it will become an increasingly important tool for building efficient and high-performing LLMs.

**Reproducibility statement.** All experiments in this paper were conducted on NVIDIA A100 (40GB) or GH200 (80GB) GPUs. We used an open-source LLM evaluation framework along with publicly available datasets and models, introducing only minor code modifications to induce activation sparsity. To ensure reproducibility, we have published our code at `https://github.com/fszatkowski/activation-sparsity-benchmarking`.

**Ethics statement.** Our work aims to advance the understanding of activation sparsity in large language models (LLMs). This phenomenon can be leveraged to obtain more efficient, robust, and interpretable models, thereby democratizing access to LLMs, reducing their cost, and making them safer. We do not identify any immediate ethical concerns specific to our method; however, as with any technique that improves the capabilities of LLMs, there is potential for misuse. We therefore urge that any deployment of resulting models be approached with caution and with consideration of broader societal impacts.

In line with the ICLR LLM policy, we disclose that LLMs were used only to refine the writing of this manuscript. All ideas and content originate from the authors.

**Contributions statement.** Filip was the primary contributor to this work, leading the development of the core ideas, the majority of the implementation, and the writing of the manuscript. Patryk contributed the code and experimental results and helped write the section on the diffusion LLMs. The remaining authors provided input on the research direction and contributed to the writing and revision of the manuscript throughout the project. Author order beyond the first author is alphabetical.

## ACKNOWLEDGEMENTS

This research was supported by National Science Centre (NCN, Poland) Grants No. 2022/45/B/ST6/02817 and 2024/53/N/ST6/03078. This research was partially funded by National Science Centre, Poland, grant no: 2023/51/D/ST6/02846. This paper has been supported by the Horizon Europe Programme (HORIZON-CL4-2022-HUMAN-02) under the project "ELIAS: European Lighthouse of AI for Sustainability", GA no. 101120237. We gratefully acknowledge the Polish high-performance computing infrastructure PLGrid (HPC Center: ACK Cyfronet AGH) for providing computer facilities and support within the computational grants no. PLG/2024/017202, PLG/2025/018230, PLG/2025/018391.

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

# Appendix

## A    HOW TO INDUCE ACTIVATION SPARSITY IN FFNS?

### A.1    ALTERNATIVE SPARSIFICATION RULES

In Section 3, we propose to use the top-p sparsification rule to induce the sparsity in the activation vectors of the models. We opt for a simple sparsification rule to avoid any data dependency or bias towards a specific model or FFN module. However, top-p is not the only possible way to perform sparsification, and many other works opted for alternative methods to extract the sparse subsets of neurons, such as top-k (Zhang et al., 2021) or max-p (Szatkowski et al., 2024).

For vector $v \in \mathbb{R}^n$, top-k finds $k$ largest neurons in the vector and can be formally defined as a transformation that multiplies $v$ with a subset of $k$ neurons which maximizes the norm of the sparsified vector:

$$\text{top-k}(v) = m_k \odot v; \ m_k = \arg\max_m ||m \odot v||_1 \ \text{ s.t. } ||m||_0 = k \text{ and } m \in \{0,1\}^n.$$

Similarly, max-p finds the subset of the neurons that satisfy the condition that their absolute values are at least $p \cdot \max(v)$:

$$\text{max-p}(v) = m_p \odot v; \ m_p = \arg\min_m ||m||_0 \ \text{ s.t. } |v_i| \cdot m_i \geq p \cdot ||v||_\infty \ \ \forall i \ \text{ and } m \in \{0,1\}^n,$$

where $||v||_\infty = \max_i |v_i|$ denotes the maximum absolute entry of $v$, and the mask $m_p$ retains exactly those coordinates $i$ for which $|v_i| \geq p \cdot ||v||_\infty$. Notably, the mask always selects the largest entry in the activation vector.

We empirically compare the three sparsification strategies in Figure 8, focusing on the sparsity–accuracy tradeoff averaged over our evaluation tasks for the smallest model in each family. Overall, top-p and top-k produce very similar curves, whereas max-p underperforms in certain settings. Therefore, we adopt top-p for our experiments, as it is more interpretable than top-k and can more universally transfer across model sizes. In particular, larger models typically yield higher sparsity, requiring $k$ to be carefully chosen as most values of $k$ have no effect until a critical sparsity is reached. By contrast, with top-p performance degrades more smoothly and predictably, allowing us to evaluate a fixed set of thresholds that transfer well across models.

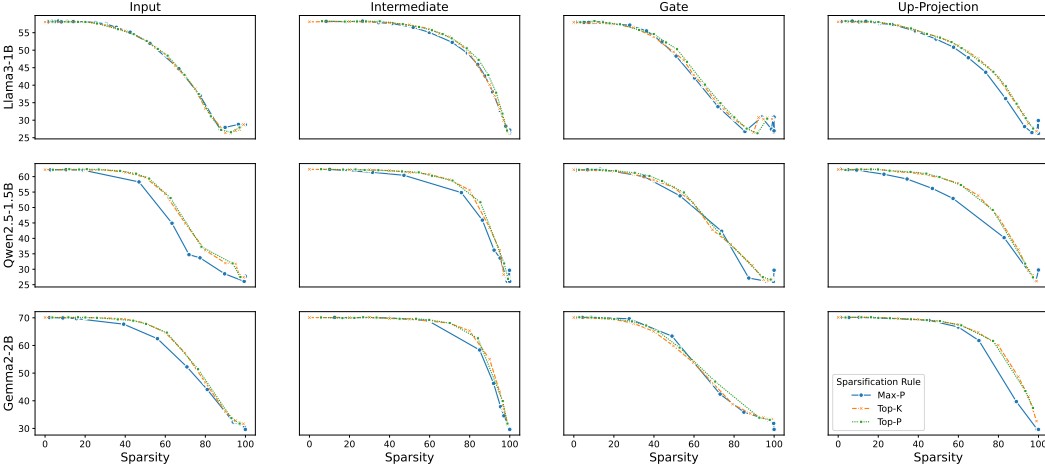

Figure 8: Comparison of sparsification rules for different models and different blocks.

## A.2 GENERALIZED top-p OPERATOR

We further investigate generalized top-p rule, denoted as top-p$^L$, which is given by:

$$\text{top-p}^L(v) = m_p \odot v; \ m_p = \arg\min_m ||m||_0 \ \text{s.t.} \ ||m \odot \hat{v}||_1 \geq p \cdot ||\hat{v}||_1 \ \text{and} \ m \in \{0,1\}^n, \quad (1)$$

where $\hat{v} = |v|^L$ for $L > 0$. This formulation potentially allows finer control over the sparsification algorithm, with $L > 1$ making the sparsification rule with a given $p$ select smaller subsets of large activations, while $L < 1$ leading to rule usually selecting more neurons for a given threshold. Notably for $L = 1$ the rule is equivalent to the top-p used in the main body of our paper.

We compare top-k, and top-p$^L$ with $L = \{0.5, 1.0, 2.0\}$ on Qwen2.5 models, applying sparsification to all-inputs, intermediate and gate activations, and present the findings in Figure 9. There are slight fluctuations in the sparsity characteristics obtained with the tested rules, but all approaches produce very similar results. While particular values of $p$ or $k$ might transfer to different sparsity levels, with sufficiently dense sampling of the thereholds the characteristics for all the rules cover similar sparsity levels, and there are no meaningful perfomance gains from using one given approach. This further supports our choice of standard top-p as the default sparsification operator, as its simply easier to practically apply, while performing as well as the other alternatives.

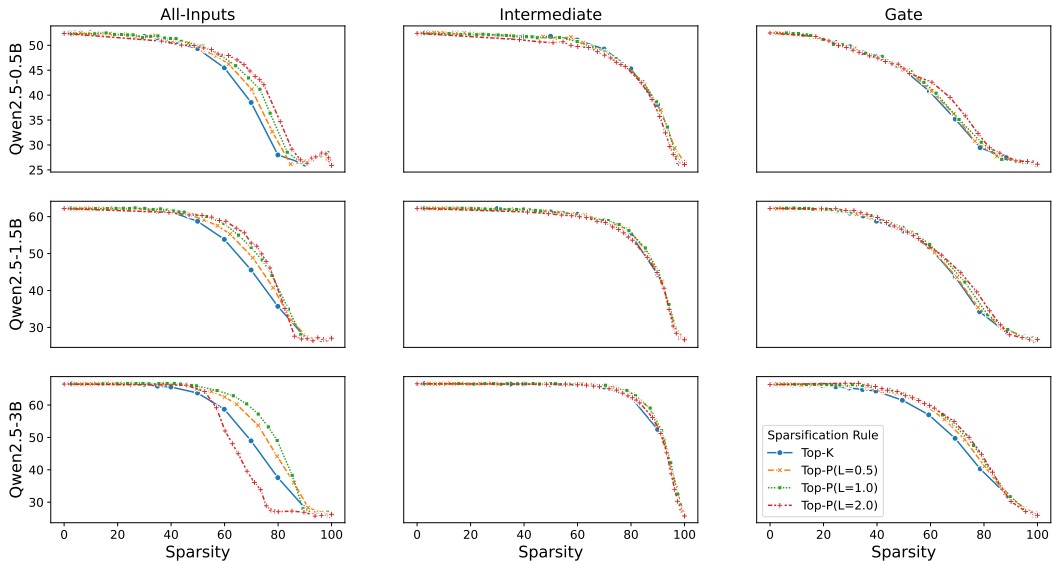

Figure 9: Accuracy characteristics under sparsification with variants of generalized top-p and top-k for Qwen2.5 0.5B, 1.5B, and 3B.

### A.3 Sparsification rule transferability between the models

To further study the transferability and impact of the threshold selection in different models, we investigate the activation sparsity induced in separate layers of Gemma3 and Qwen2.5 models. We select a subset of $p$ thresholds, and register the activation sparsity obtained at a given layer alongside the average of the accuracy under the threshold. We plot the results as heatmaps in Figures 10 and 11.

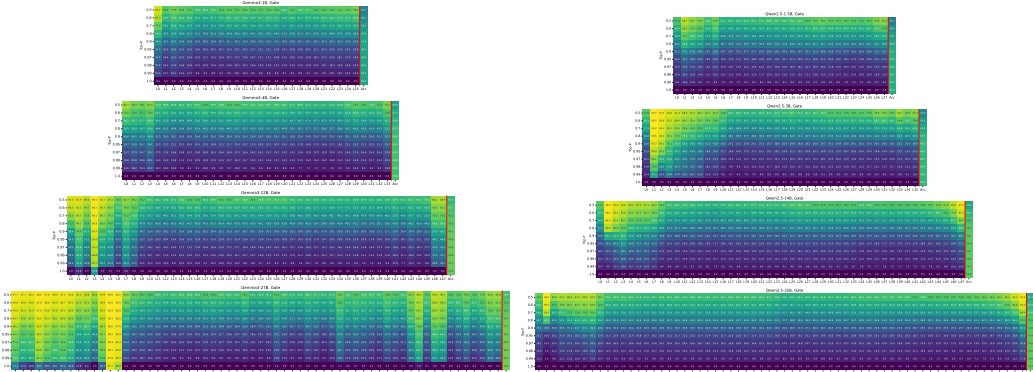

Figure 10: Top-p threshold values and resulting sparsity induced in the gate activation vectors alongside the accuracy with the given threshold across different layers of Gemma3 and Qwen2.5.

First, we investigate the sparsity of gate activations in the Gemma and Qwen models of corresponding sizes in Figure 10. Except for some early layers, the sparsity values obtained across the models appear similar for a given threshold across the middle layers.

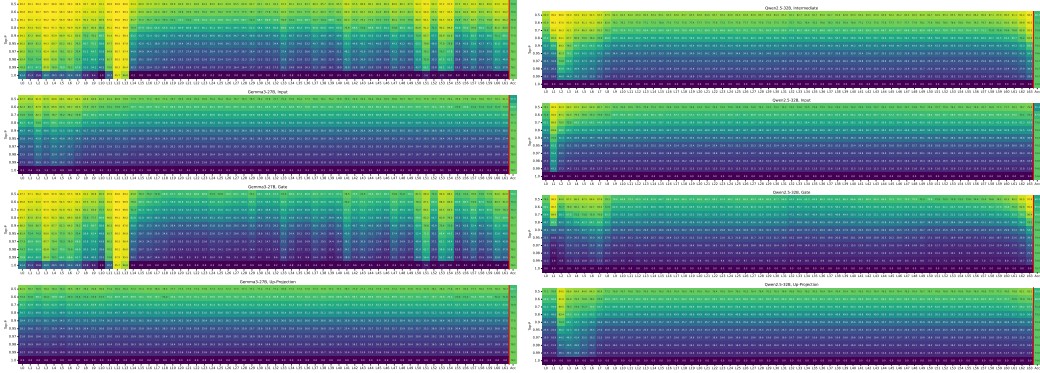

Figure 11: Top-p threshold values and resulting sparsity for Gemma3-27B and Qwen2.5-32B models.

Then, in Figure 10 we plot the sparsity of all four investigated activation vector types in the largest models within each model family. Again, except for a few layers, the sparsities obtained for a given $p$ appear similar between the models.

Both of these results support the universality of our approach and our decision to choose top-p over top-k, as using top-k would require more manual threshold selection to find the critical sparsity, as outlined by the variance in the critical sparsity across different model sizes and types in Section 4. While the resulting sparsity heatmaps show a few high outliers, particularly for Gemma3-27B gates, we attribute the presence of these to the presence of massive activations (Sun et al., 2024), as for massive outlier values, the magnitude of the vector norms that we use will concentrate around very few large values and may even cause exact sparsity to appear at $p = 1.0$ as the nature of the numerical precision will make the smallest entries in the activation vector appear like zeros since they basically contribute nothing compared to the massive outlier. We do not investigate this phenomenon further and leave it for future work. However, we note that it can have important implications for the design of activation sparsity approaches, particularly those that rely on thresholding, as rules and thresholds devised for massive activations might be highly unstable upon encountering out-of-distribution data.

# B    CRITICAL ACTIVATION SPARSITY OF PRETRAINED AND INSTRUCTION-TUNED MODELS

Table 2: Critical activation sparsity for pretrained and instruction-tuned models. $S_{inter}$, $S_{all\_in}$, $S_{gate}$, $S_{input}$ and $S_{up\_p}$ refer to intermediate, all inputs, gate, input and up-projection sparsification, respectively, and $L$, $d_m$ and $d_i$ stand for number of layers, model and intermediate dimensionality.

| Model | $L$ | $d_m$ | $d_i$ | Pretrained | | | | | Instruction-Tuned | | | | |
| | | | | $S_{inter}$ | $S_{all\_in}$ | $S_{gate}$ | $S_{input}$ | $S_{up_p}$ | $S_{inter}$ | $S_{all\_in}$ | $S_{gate}$ | $S_{input}$ | $S_{up_p}$ |
|---|---|---|---|---|---|---|---|---|---|---|---|---|---|
| Gemma3-1B | 26 | 1152 | 6912 | 50.22 | 35.15 | 22.83 | 28.53 | 32.96 | 49.98 | 35.89 | 23.23 | 32.14 | 30.27 |
| Gemma3-4B | 34 | 2560 | 10240 | 58.56 | 40.46 | 28.50 | 34.63 | 39.72 | 62.82 | 47.78 | 35.99 | 42.29 | 40.82 |
| Gemma3-12B | 48 | 3840 | 15360 | 69.46 | 52.54 | 42.05 | 43.03 | 42.03 | 78.77 | 61.45 | 55.45 | 50.74 | 54.26 |
| Gemma3-27B | 62 | 5376 | 21504 | 74.12 | 66.26 | 53.03 | 50.83 | 42.01 | 84.05 | 74.25 | 68.15 | 59.88 | 56.95 |
| LLaMA3.2-1B | 16 | 2048 | 8192 | 44.44 | 30.93 | 26.51 | 28.09 | 28.82 | 45.02 | 32.52 | 24.30 | 29.65 | 29.70 |
| LLaMA3.2-3B | 28 | 3072 | 8192 | 49.58 | 34.51 | 25.52 | 35.91 | 37.14 | 58.07 | 36.62 | 28.90 | 33.28 | 44.72 |
| LLaMA3.1-8B | 32 | 4096 | 14336 | 51.89 | 38.97 | 28.04 | 37.31 | 30.52 | 61.96 | 44.35 | 34.38 | 39.34 | 41.76 |
| Qwen2.5-0.5B | 24 | 896 | 4864 | 46.54 | 36.66 | 17.16 | 42.01 | 29.20 | 43.92 | 33.20 | 24.60 | 32.80 | 32.12 |
| Qwen2.5-1.5B | 28 | 1536 | 8960 | 50.49 | 40.98 | 25.93 | 40.12 | 35.50 | 52.93 | 39.14 | 27.63 | 32.50 | 32.99 |
| Qwen2.5-3B | 36 | 2048 | 11008 | 71.16 | 50.43 | 39.58 | 39.40 | 43.46 | 59.80 | 47.47 | 36.90 | 44.16 | 36.32 |
| Qwen2.5-7B | 28 | 3584 | 18944 | 60.98 | 51.77 | 37.25 | 47.89 | 43.05 | 59.95 | 51.73 | 32.58 | 47.01 | 40.62 |
| Qwen2.5-14B | 48 | 5120 | 13824 | 71.66 | 54.66 | 48.04 | 47.39 | 52.25 | 69.35 | 56.89 | 41.87 | 50.10 | 49.04 |
| Qwen2.5-32B | 64 | 5120 | 27648 | 65.66 | 58.93 | 40.20 | 54.08 | 52.46 | 68.77 | 62.83 | 40.54 | 55.17 | 57.35 |

In Table 2, we report the exact numerical values of the critical activation sparsity for all models considered in our experiments, including both pretrained and instruction-tuned variants. For context, we also include key model architectural parameters such as the number of layers, model dimensionality, and intermediate dimensionality. While we do not observe clear, direct relationships between these parameters and the achieved critical sparsity, the general trend of sparsity increasing with model size remains evident. Notably, the Qwen family exhibits some fluctuations, which may partially be explained by the non-uniform scaling of its architectural parameters across model sizes.

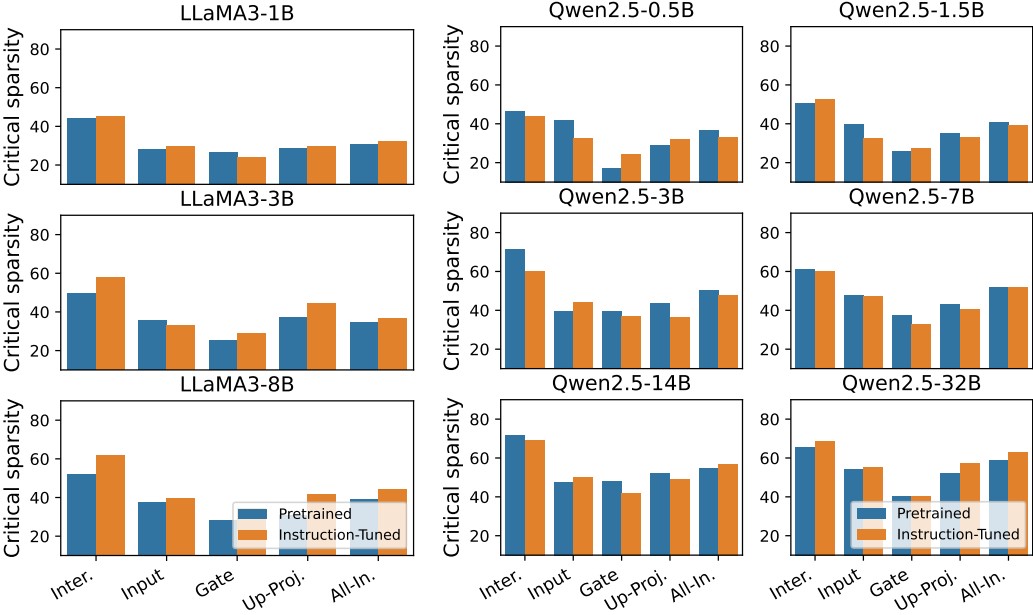

Figure 12: Critical sparsity plots for LLaMA and Qwen models, corresponding to the Figure 5a.

# C  AVERAGE ACCURACY OF PRETRAINED QWEN2.5 AND LLAMA3 MODELS.

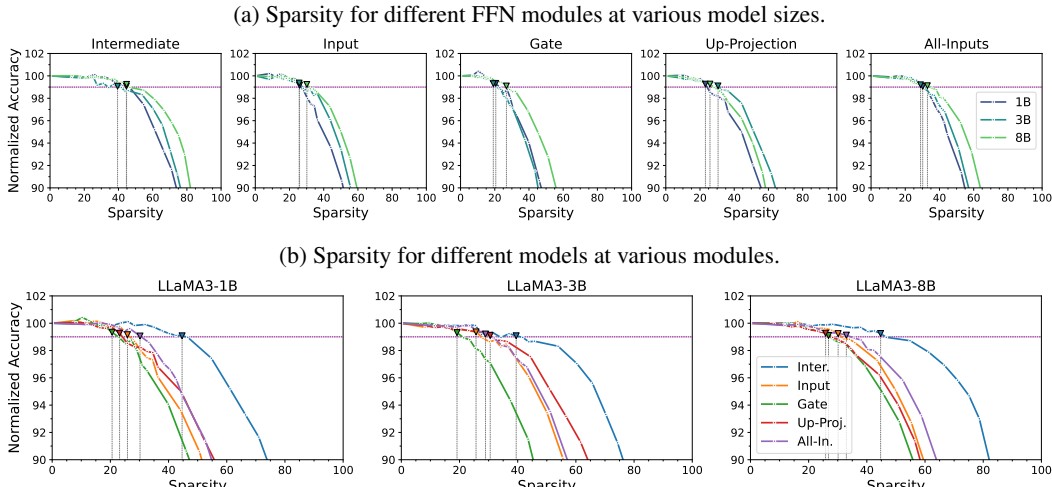

Figure 13: Average accuracy across downstream tasks normalized by the original performance with different induced activation sparsity for base LLaMA3 models, corresponding to the plots in Figure 2.

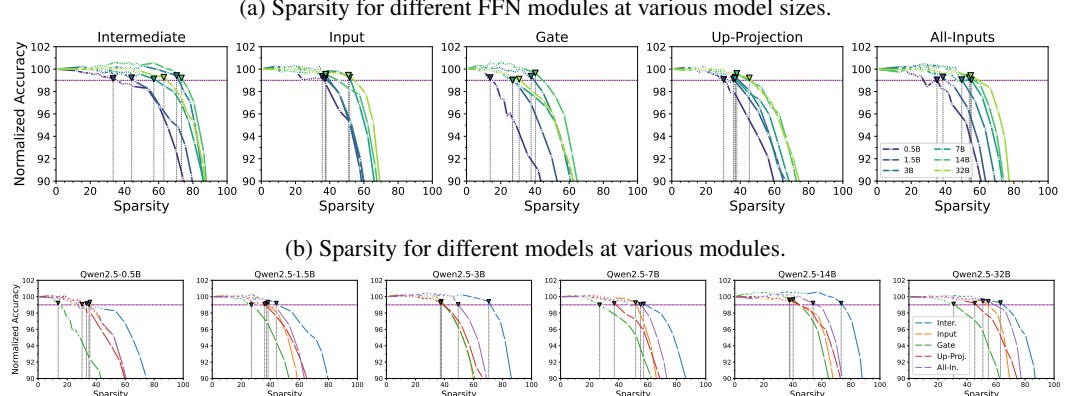

Figure 14: Average accuracy across downstream tasks normalized by the original performance with different induced activation sparsity for base Qwen2.5 models, corresponding to the plots in Figure 2.

# D    PER-TASK SPARSITY FOR LLAMA AND QWEN MODELS

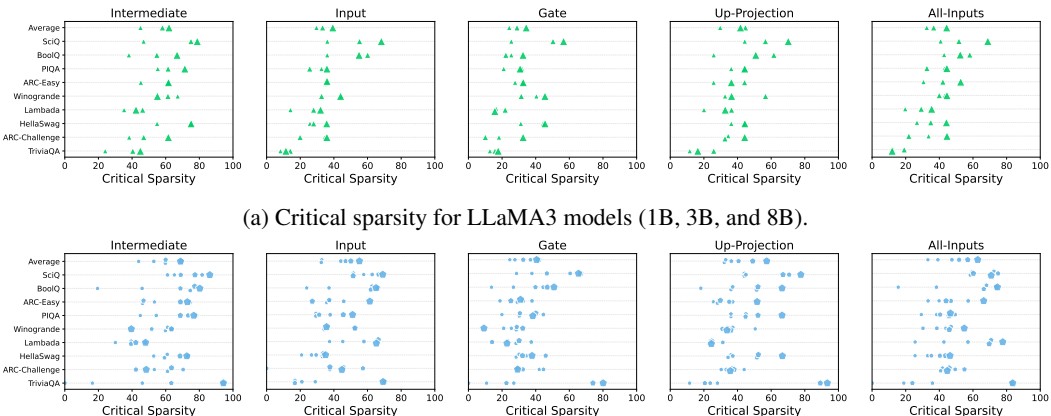

(a) Critical sparsity for LLaMA3 models (1B, 3B, and 8B).

(b) Critical sparsity for Qwen2.5 models (0.5B, 1.5B, 3B, 7B, 14B, and 32B).

Figure 15: Critical sparsity of different tasks across all the evaluated modules. The marker size denotes the model size, and the tasks are ordered by their accuracy (the tasks at the top achieve the highest accuracy). The sparsity at which the model performance degrades varies considerably between the tasks. Although the "easier" tasks with higher accuracy tend to be more tolerant to sparsity, there is no clear correlation, which suggests that activation sparsity patterns tend to be highly data-dependent.

# E    EFFECTIVE RANK COMPUTATION

To compute effective ranks of activations in Section 4.2, we follow the formula from (Roy & Vetterli, 2007), which we restate here for the sake of the paper's self-containment. In our experiments, we apply the following formula to each batch and then report the effective rank as the average across all the batches.

To compute the effective rank of activation matrix $A \in R^{n \times d}$, where $n$ refers to the number of tokens within a batch, and $d$ is the feature dimensionality of the model, we first compute singular value decomposition (SVD) of $A$:

$$A = UDV,$$

where $D \in R^{n \times d}$ is a diagonal matrix containing the singular values $\sigma_1 \geq \sigma_2 \geq \cdots \geq \sigma_Q \geq 0$, and $Q$ is defined as $Q = \min(n, d)$.

We can further define $\sigma = (\sigma_1, \sigma_2, \ldots, \sigma_Q)^T$, and the singular value distribution is then given by:

$$p_k = \frac{\sigma_k}{||\sigma||_1} \quad \text{for} \ \ k = 1, 2, \ldots, Q,$$

where $^T$ denotes transposition and $|| \cdot ||_1$ is the L1 norm.

The effective rank of $A$ is then defined:

$$\text{erank}(A) = \exp\{H(p_1, p_2, \ldots, p_Q)\},$$

where $H(p_1, p_2, \ldots, p_Q)$ is the Shannon entropy of the singular value distribution.

# F    ADDITIONAL RESULTS FOR MoE MODELS

## F.1    INTER-TASK VARIANCE IN CRITICAL SPARSITY LEVELS FOR QWEN3-30B-A3B

In this section, we present the results complementary to the ones presented in Section 4.5. Similar to Figure 6a, we investigate minimum, maximum, and average sparsity in each layer in Qwen3-30B-A3B across the isolated tasks from our task suite. Interestingly, we find that average sparsity is very similar across each task, and the only notable variance in the model behaviour can be observed in the maximum sparsity, which sometimes fluctuates per expert. While our previous results for the dense models indicated a large variance in critical sparsity per task, these results rather indicate that such variances average out across large numbers of experts in MoE architectures.

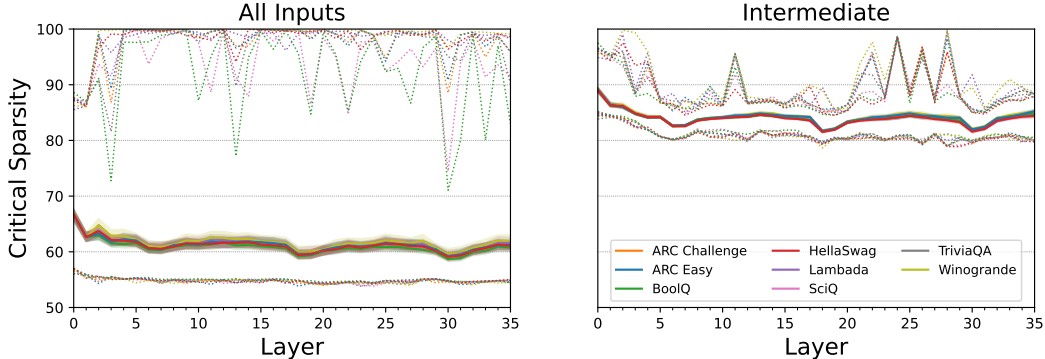

Figure 16: Critical sparsity across Qwen3-30B-A3B layers measured per-task.

## F.2 RESULTS FOR OLMoE-7B-A1B

To complement our analysis, we repeat Qwen3-30B-A3B experiments with smaller MoE, OLMoE-7B-A1B (Muennighoff et al., 2025). OLMoE has 16 MoE layers and uses 8 out of 64 experts for a given token. In Figures 17 and 18 we present the plots corresponding to the Figure 6a from the main paper and the per-task activation sparsity analysis presented in Figure 16. Overall, the sparsity in the 7B-A1B model is smaller than in the 30B-A3B Qwen, and the per-expert variance is smaller, as also expected. Interestingly, in this model, intermediate sparsity is close to all-inputs, which can be explained by the lower expansion rate in OLMoE FFN, where the intermediate dimension is just two times larger than the hidden dimension. Nonetheless, the interesting property of experts showing small variance in sparsity across the tasks observed for the Qwen model persists in the smaller MoE.

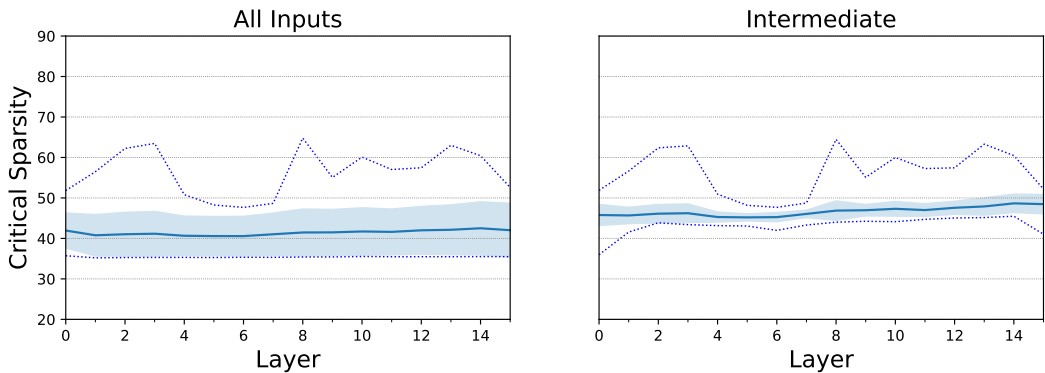

Figure 17: Critical sparsity across OLMoE-7B-A1B across the layers, with standard deviation shaded and the minimum and maximum sparsity for each expert marked with dotted lines.

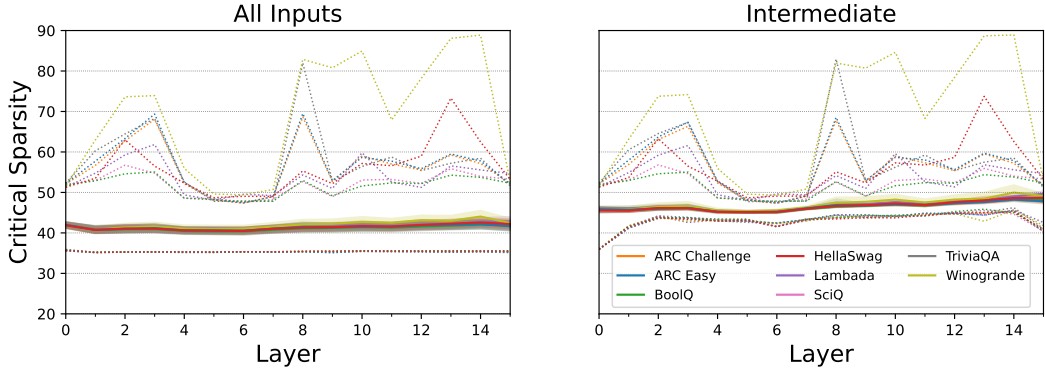

Figure 18: Critical sparsity across OLMoE-7B-A1B layers measured per-task.

