# OpenReview forum: "Universal Properties of Activation Sparsity in Modern Large Language Models"
_ICLR.cc/2026/Conference — ICLR 2026 Poster_

### Official Review · Reviewer_BibT · 2025-10-27

**Soundness:** 3
**Presentation:** 3
**Contribution:** 3
**Rating:** 6
**Confidence:** 4

**Summary:**

This paper presents a comprehensive study on the properties of sparsification in the FFN layers of transformer-based language models and diffusion language models. Specifically, they use a simple top-$p$ sparsification strategy and analyze the robustness against sparsity of various models from the Gemma3, Qwen2.5, and LLaMa3.1/3.2 family, by measuring at what sparsity levels does downstream task performance drop below 99% of its original performance.

Some interesting findings from this paper include: input-based sparsification is often more attractive than gate-based sparsification, larger models are more robust to sparsification, and diffusion models are more robust to sparsification. The findings of this paper are valuable for designing language models with better robustness against sparsification or methods that better leverage sparsification to achieve inference speedup.

**Strengths:**

- The paper is well-presented and generally easy to follow.
- The findings of this paper is very interesting and highly valuable to researchers interesting in language models with sparse activations.
- The methods and experimental setup is well designed, making the results reliable and convincing.

**Weaknesses:**

- How sparse activation patterns evolve across diffusion steps is an interesting phenomenon and the paper does a good job in bringing it up (starting with Line 406, and Figure 6). However, the conclusions of the experiments in this regard is inconclusive and it is unclear how they are useful for the community.
- Considering how sparse mixture-of-experts (MoE) models have become very popular, I would love to see investigations in how the findings of this paper transfers to MoE models. I suggest repeating some of the experiments with models such as MoE variants of the Qwen3 series, GPT-OSS, GLM4.5, etc.
- While the top-$p$ sparsification strategy is reasonable, I would like to see a comparison of it against an even simpler, top-$k$ sparsification strategy.

**Questions:**

- What does "Normalized accuracy" in Figure 5 mean?
- Can you provide more insights into why MMLU-Redux is less robust to sparsification compared to GSM8K and TruthfulQA?
- It seems that you apply the same $p$ value across all layers. However, I think using a different $p$ for different layer is more reasonable. Is it possible to repeat the experiments, but with a layer-specific $p$ value such that each layer can have its own critical sparsity?

---

> ### Author Response · Authors · 2025-11-21
> **Rebuttal response (1/2)**
>
> We sincerely thank the Reviewer for their careful reading of our manuscript and for providing valuable suggestions to further strengthen our work. We are pleased that the reviewer views the comprehensive evaluation we conducted and the findings derived from our experiments as interesting for the community. Below, we address each of the Reviewer’s points in detail.
>
> > How sparse activation patterns evolve across diffusion steps is an interesting phenomenon and the paper does a good job in bringing it up (starting with Line 406, and Figure 6). However, the conclusions of the experiments in this regard is inconclusive and it is unclear how they are useful for the community.
>
> **Our goal in this experiment was to assess whether activation sparsity masks in diffusion-based LLMs remain consistent across diffusion steps**, potentially enabling acceleration techniques similar to [1]. Prior internal results showed that per-step sparsity remains high throughout the denoising process, which prompted us to investigate how the underlying sparsity patterns evolve and whether they exhibit consistency across steps. Our findings, however, indicate that no meaningful, consistent patterns emerge.
>
> In response to the Reviewer’s comment, **we have updated Section 4.6 to include the previously omitted per-step sparsity results and to present a clearer motivation for the experiments**. We hope the updates have improved the overall readability of the section. We also remain open to further suggestions on how to enhance the section if the Reviewer finds it necessary.
>
> > Considering how sparse mixture-of-experts (MoE) models have become very popular, I would love to see investigations in how the findings of this paper transfers to MoE models. I suggest repeating some of the experiments with models such as MoE variants of the Qwen3 series, GPT-OSS, GLM4.5, etc.
>
> We thank the Reviewer for the suggestion and agree that testing whether our findings extend to sparse MoE architectures is important given their growing role in practice. In response, **we added a study on the Qwen3-30B-A3B MoE model in Section 4.5, analyzing its activation sparsity and per-expert variability**. These results demonstrate that activation sparsity is also clearly present in MoE models, and interestingly highlight that a few experts display unusually high sparsity, even though the overall variance across experts, as measured by the standard deviation, remains modest.
>
> Currently, we report the results for a subset of 6 out of the total 9 tasks in our evaluation suite. The remaining experiments are already scheduled, and we will update the manuscript shortly once these experiments finish and possibly provide further results for the MoEs within our computational budget. We hope this addresses the Reviewer’s request and remain open to further discussion.
>
> > While the top- sparsification strategy is reasonable, I would like to see a comparison of it against an even simpler, top-k sparsification strategy.
>
> We have included a brief comparison of top-k, top-p, and max-p sparsification in Appendix A1 in the original paper. Since top-k and top-p behave nearly identically, we chose top-p for its simplicity and clearer interpretability. In response to Reviewer 7CAX, we have also expanded this analysis to cover a generalized top-p rule in the new Appendix A2. These results confirm that **the standard top-p variant offers the best balance of practicality and performance** for our purposes.
>
> > What does "Normalized accuracy" in Figure 5 mean?
>
> We normalize the accuracy values of the sparsified model by the corresponding accuracy of the dense model. This was mentioned in the original manuscript, e.g., in the caption to Figure 2: “ We normalize the accuracy by the original performance of the dense models”. This normalization allows for a clear comparison of performance degradation across different models and tasks, which would otherwise start from different dense accuracies at zero sparsity.
>
> However, we are grateful to the reviewer for highlighting that the term might not be clear everywhere we use it. **We have modified the manuscript to clearly define the term and eliminate any potential confusion** and updated the start of the experiments section and caption to Figure 5.
>
> Please refer to the second comment for the answers to the remaining issues and references.

---

> ### Author Response · Authors · 2025-11-21
> **Rebuttal response (2/2)**
>
> > Can you provide more insights into why MMLU-Redux is less robust to sparsification compared to GSM8K and TruthfulQA?
>
> It is not an issue specific to MMLU-Redux, but to the reasoning models. Because of computational limits, we evaluated all models with a fixed output length. For reasoning models, **MMLU responses are typically longer than those for the other tasks, and when we impose additional activation sparsity, the model sometimes fails to produce an “end of thinking” token within the allowed output token length**. As a result, the sparsified reasoning model sometimes never outputs an answer, which shows up as degraded MMLU performance. We expect this effect to diminish if the output length were unconstrained, but such an experiment currently exceeds our available computational resources.
>
> > It seems that you apply the same  value across all layers. However, I think using a different  for different layer is more reasonable. Is it possible to repeat the experiments, but with a layer-specific  value such that each layer can have its own critical sparsity?
>
> We thank the reviewer for this insightful comment. Indeed, several prior works referenced in our paper (e.g., CATS [2], TEAL [3], LaRoSa [4]) determine sparsity thresholds on a per-layer basis, and our approach could in principle also use different $p$ values on a layer-wise basis. While such calibration could lead to higher functional sparsity, it also introduces significant computational overhead to find the critical sparsity levels, as one must optimize hyperparameters over exponentially more possible combinations. In practice, **this requires additional assumptions and dedicated optimization procedures, which run counter to our main design goal of keeping the sparsification rule simple and practical**.
>
> Importantly, even with a single global top-$p$ value, the resulting sparsity is not uniform across layers. As shown in the current Appendix A3, the same top-$p$ threshold naturally induces different activation sparsity per-layer. The reported critical sparsity values in the paper are therefore averaged over layers for a given global top-$p$. Layer-wise critical sparsities can be directly inferred from the plots in Appendix A3 by identifying, for each layer, the lowest top-$p$ that maintains a set accuracy constraint.
>
> ### Conclusion
>
> We thank the Reviewer for their careful reading and constructive feedback, which has helped us improve the paper. We hope our detailed responses, updated experiments, and clarifications have sufficiently demonstrated the rigor, novelty, and practical value of our work. Given the Reviewer’s initial acknowledgment of the quality of our work, we hope these revisions further improve their assessment of our work.
>
> ### References
>
> * [1] *Prompt-prompted Adaptive Structured Pruning for Efficient LLM Generation.*, Dong et al., COLM 2024.
>
> * [2] *CATS: Context-Aware Thresholding for Sparsity in Large Language Models*, Lee et al., COLM 2024.
>
> * [3] *Training-Free Activation Sparsity in Large Language Models.*, Liu et al., ICLR 2025.
>
> * [4] *La RoSA: Enhancing LLM Efficiency via Layerwise Rotated Sparse Activation.*, Liu et al., ICML 2025.

---

> ### Comment · Reviewer_BibT · 2025-11-26
>
> Thank you for the detailed response and clarifications. Some of my concerns have been resolved. Thus, I have decided to increase the presentation score.

---

### Official Review · Reviewer_PkEJ · 2025-10-31

**Soundness:** 3
**Presentation:** 3
**Contribution:** 3
**Rating:** 4
**Confidence:** 5

**Summary:**

This work proposes a unified framework for evaluating sparsity robustness in large language models and systematically examines activation sparsity in their feedforward layers. The study identifies consistent sparsity patterns across different model families and sizes, revealing that larger models exhibit greater potential for effective activation sparsity. It also provides the first analysis of sparsity in diffusion-based LLMs. Overall, the work offers insights and guidance for leveraging sparsity to improve LLM efficiency and design.

**Strengths:**

- The paper uncovers some universal properties of activation sparsity across diverse model families and scales. This could be quite helpful to the community.
- The paper is well written.
- Experiments are quite quantitive and informative.

**Weaknesses:**

- Pure experimentation-driven work without any theoretical analysis or others.
- Larger model leads to higher critical sparsity. This point is actually very straightforward. For instance, usually larger model and smaller model are trained on the same pile of data. Of course the larger model admit more redandancy.

**Questions:**

- What is  "functional sparsity"? The authors not seem to provide good explanation across the paper, though this term is very important.
- What is "effective rank"? Need short explanation in the paper to be self-contained. For instance, what "effective rank of 0.1" means? How it corresponds to your conclusion.

Personally, I like this type of paper giving inspiration. Please work on the quesitons and weakness points if appropriate.

---

> ### Author Response · Authors · 2025-11-21
> **Rebuttal response**
>
> We thank the Reviewer for the insightful feedback and encouraging comments. We are happy that the Reviewer praises our writing and sees the potential value of our study for the community. Below, we address and clarify the raised concerns.
>
> > Pure experimentation-driven work without any theoretical analysis or others.
>
> While we agree that our work is primarily empirical, we stress that this is intentional. Existing theoretical analyses of activation sparsity focus on ReLU networks and do not apply to modern GLU/SiLU/GELU-based LLMs. **We believe that before such a theory can be developed, a unified empirical characterization is necessary.** Our contribution is to provide a systematic, model-agnostic framework for measuring activation sparsity in LLMs, reveal consistent cross-model patterns, and demonstrate empirical regularities that future analysis can build upon.
>
> Importantly, other Reviewers (7CAX, YvVz, BibT) also mentioned our comprehensive experiments as a strength of our submission and highlighted their practical value for the community. Therefore, **we believe our work stands on its own as a solid and relevant contribution to ICLR and can serve as a useful foundation for deeper theoretical work in the future**.
>
> > Larger model leads to higher critical sparsity. This point is actually very straightforward. For instance, usually larger model and smaller model are trained on the same pile of data. Of course the larger model admit more redandancy.
>
> While we agree with the general intuition of the reviewer, in practice, this trend does not always hold, and **redundancy does not automatically translate to effective sparsity**. Despite the intuitive expectation that more parameters would lead to more redundancy and integrate more smoothly with such techniques, **scaling LLM compute or model size does not always translate directly to better model robustness**, as shown in works on quantization [1,2,3] or pruning [4]. Moreover, certain phenomena, such as massive activations [5] or attention sinks [6], emerge only at larger scales, which can also lead to unexpected behaviors.
>
> Our work aims to quantify scaling effects and empirically spot such non-obvious trends. For instance, we observe that FFN gate activations are less robust to sparsity than plain input activations, which seems counterintuitive, as gate activation vectors affected by activation functions should be sparser. We also capture scaling trends such as gate sparsity robustness increasing more rapidly with model size than input, which is not trivial to anticipate without empirical evaluation.
>
> While some results may seem intuitive to the Reviewer, they are not necessarily obvious to every reader, and the **detailed trends and quantitative measurements in our paper provide valuable guidance for practitioners**. Positive reception from other Reviewers (7CAX, YvVz, BibT) further underscores the utility of these contributions.
>
> > What is "functional sparsity"? The authors not seem to provide good explanation across the paper, though this term is very important.
>
> We use “functional sparsity” to refer to the practical activation sparsity that can be imposed on a model without significantly degrading its performance. This concept is particularly important for evaluating activation sparsity in architectures that do not produce exact zeros (e.g., models without ReLU). We acknowledge that the term was previously used somewhat loosely, under the assumption that it would be intuitively understood.
>
> In response to the reviewer’s comment, we have updated the paper introduction and Section 3 to clarify this concept and avoid ambiguity. We welcome any additional suggestions the reviewer may have for improving the readability further.
>
> > What is "effective rank"? Need short explanation in the paper to be self-contained. For instance, what "effective rank of 0.1" means? How it corresponds to your conclusion.
>
> We use effective rank as defined in [7]. In our plots, we normalize the rank by the activation dimensionality to make it comparable across different model sizes and activation types, since the same absolute rank can have very different implications depending on the full vector dimensionality.
>
> We thank the reviewer for pointing out this potential ambiguity. **We have added Appendix E detailing the effective rank computation and updated the relevant section in the paper to reference it.**
>
> ### Conclusion
>
> We thank the Reviewer once again for the time and effort spent on our paper. We have updated the paper according to the suggestions, and hope to have addressed the remaining concerns in our answer. Given the positive feedback from the other Reviewers regarding the novelty (YvVz), rigor (7CAX, BibT), and practical value of our work (7CAX, YvVz, BibT), we hope that our response convinces the Reviewer about the contribution and relevance of our paper. We would greatly appreciate it if the Reviewer would reconsider the positive assessment of our work.

---

> ### Author Response · Authors · 2025-11-21
>
> ### References
>
> * [1] *Exploring the Trade-Offs: Quantization Methods, Task Difficulty, and Model Size in Large Language Models From Edge to Giant.*, Lee et al., IJCAI 2025.
>
> * [2] *When Quantization Affects Confidence of Large Language Models?*, Proskurina et al., NAACL 2024.
>
> * [3] *Scaling Laws for Precision.*, Kumar et al., ICLR 2024.
>
> * [4] *BlockPruner: Fine-grained Pruning for Large Language Models”, Zhong et al., ACL 2025
>
> * [5] *Massive Activations in Large Language Models.*, Sun et al., COLM 2024.
>
> * [6] *When Attention Sink Emerges in Language Models: An Empirical View*, Gu et al., ICLR 2025
>
> * [7] *The Effective Rank: a Measure of Effective Dimensionality.*, Roy and Vetterli, EUSIPCO 2007.

---

> > ### Comment · Reviewer_PkEJ · 2025-11-27
> > **Thank you for the response**
> >
> > The concerns are addressed, so I change the raiting to 6.

---

### Official Review · Reviewer_YvVz · 2025-11-01

**Soundness:** 3
**Presentation:** 3
**Contribution:** 3
**Rating:** 6
**Confidence:** 3

**Summary:**

The paper addresses the lack of a general understanding of activation sparsity in modern Large Language Models (LLMs) that utilize non-ReLU activations, which prohibits the use of traditional zero-based sparsity methods. To solve this, the authors introduce a novel, general framework to evaluate sparsity robustness in contemporary LLMs, focusing on the Feedforward Network (FFN) layers. Through systematic investigation across diverse models and scales, the work reveals a key universal property: the potential for effective activation sparsity increases with model size. Furthermore, the study presents the first analysis of activation sparsity in diffusion-based LLMs, ultimately offering a comprehensive perspective and practical guidance for leveraging this phenomenon in LLM design and acceleration.

**Strengths:**

1. The paper introduces a simple, training-free top-p sparsification method and the metric of critical sparsity (maximum sparsity retaining 99\% performance), providing a unified and fair way to compare sparsity tolerance across different LLM architectures and FFN components.

2. The work systematically confirms that the critical sparsity increases with model size. This is consistently reinforced by the finding that the effective rank of activations decreases with model size, offering strong evidence that larger models inherently possess greater exploitable redundancy.

3. The investigation into diffusion-based LLMs (LLaDA) is novel, revealing that they exhibit substantial activation sparsity and even slightly more favorable sparsity-performance trade-offs than autoregressive models, highlighting a new acceleration opportunity.

4. Based on empirical data, the authors provide the practical insight that input activation sparsification is the most effective training-free approach, as its sparsity tolerance is comparable to or greater than gate or up-projection activations.

**Weaknesses:**

1. A major finding is that the critical sparsity varies substantially across different downstream tasks and training recipes (e.g., instruction-tuning). This challenges the core assumption of many prior acceleration methods that sparsification rules calibrated on an auxiliary dataset will generalize universally without overfitting

2. The paper acknowledges that the effective speedups from activation sparsity methods are practically limited to a factor of 1.3x to 1.5x, which is less compelling when compared to alternative lossless techniques like speculative decoding that can achieve up to 4x speedups.

3. The analysis is explicitly constrained to only the FFN layers, intentionally excluding the Multi-Head Attention (MHA) module. While a cost justification is provided, this limits the comprehensiveness of the "universal properties" claim within the entire Transformer architecture.

4. The use of effective rank as a theoretical proxy for redundancy is weakened by the observation that gate activations show a high effective rank yet exhibit a low empirical capacity for sparsification, suggesting that this metric is insufficient to fully capture robustness to sparsification

**Questions:**

none

---

> ### Author Response · Authors · 2025-11-21
> **Rebuttal response (1/2)**
>
> We thank the Reviewer for the time and effort dedicated to evaluating our paper, and appreciate the valuable comments provided. We are pleased to see the Reviewer's positive remarks regarding the practical contributions of our work, the novelty of our experiments, and the universality of the proposed framework. Below, we address each point in detail.
>
> > A major finding is that the critical sparsity varies substantially across different downstream tasks and training recipes (e.g., instruction-tuning). This challenges the core assumption of many prior acceleration methods that sparsification rules calibrated on an auxiliary dataset will generalize universally without overfitting
>
> Methods that rely on activation sparsity for acceleration typically assume that thresholds tuned on auxiliary, often language-modeling, datasets transfer cleanly to downstream tasks. However, the empirical results in those same papers show that this transfer is inconsistent between tasks when examined closely. **Our findings align with these observations, but we explicitly highlight this phenomenon to raise awareness in the community** (see the second response for the detailed discussion).
>
> We want to emphasize that **our intent is not to criticize existing methods**, but to show where the assumptions underlying standard sparsification techniques diverge from how models actually behave. Practical acceleration methods necessarily depend on simplifying assumptions, and these approaches do succeed in inducing activation sparsity in practical applications.
>
> > The paper acknowledges that the effective speedups from activation sparsity methods are practically limited to a factor of 1.3x to 1.5x, which is less compelling when compared to alternative lossless techniques like speculative decoding that can achieve up to 4x speedups.
>
> **We view such techniques as complementary.** It is possible to apply activation sparsity-based acceleration to a model deployed in a speculative decoding pipeline or with quantization. Therefore, we strongly believe that research on activation sparsity is still relevant to enable more efficient inference, and recent models designed specifically for efficiency already leverage activation sparsity combined with other acceleration mechanisms (see Gemma3n [4] - “(...) there are many new additions in this model, including (...) **Activation Sparsity with Statistical Top-k**.”).
>
> > The analysis is explicitly constrained to only the FFN layers, intentionally excluding the Multi-Head Attention (MHA) module. While a cost justification is provided, this limits the comprehensiveness of the "universal properties" claim within the entire Transformer architecture.
>
> Activation sparsity has been examined primarily in FFNs, as focusing on these layers captures the central behavior associated with the phenomenon in Transformer models. Given this and the practical relevance of FFN sparsification, we concentrated our analysis on FFNs.
>
> In the updated paper, we have highlighted the part of the conclusions where we argue why activation sparsity is not well-suited for MHA acceleration and why we focused on a thorough study of FFN activation sparsity as both more tractable and practically relevant. Summarizing the main points here:
>
> 1. FFNs dominate the computational cost in Transformers, especially at short context lengths.
>
> 2. FFNs have higher dimensionality, which enables greater potential sparsity, e.g., our results show that intermediate vectors can sustain higher critical sparsity than input vectors.
>
> 3. Obtaining practical acceleration gains from activation sparsity in MHA is technically challenging. Attention implementations such as FlashAttention are already highly optimized, rely on fused kernels, and leave little room for dynamic computation, making real deployment speedups infeasible.
>
> Please see the second comment for the discussion on the other points raised and the references mentioned in our answer.

---

> ### Author Response · Authors · 2025-11-21
> **Rebuttal response (2/2)**
>
> > The use of effective rank as a theoretical proxy for redundancy is weakened by the observation that gate activations show a high effective rank yet exhibit a low empirical capacity for sparsification, suggesting that this metric is insufficient to fully capture robustness to sparsification
>
> Effective rank is commonly used as a proxy for redundancy in weights and activations [5,6], so examining its relationship to activation sparsity seemed reasonable to us. Our experiments interestingly show that input activations have much higher effective rank than intermediate or gate activations. Yet, intermediate vectors are empirically the most robust to sparsification, and inputs typically sparsify better than gates. This contradicts what would be our naive expectation, that the effective rank of activation matrices would correlate with their capacity for activation sparsity.
>
> While there is still correlation between general trends (e.g. effective rank getting lower, sparsity getting higher with model size), we ourselves acknowledge that our results “suggest that effective rank alone may be insufficient to fully capture robustness to sparsification“. **We report these findings to inform practitioners seeking proxies for activation sparsity**, as they highlight that the intuitive metric of effective rank does not reliably predict robustness to sparsification.
>
> > (... the paper) challenges the core assumption of many prior acceleration methods that sparsification rules calibrated on an auxiliary dataset will generalize universally without overfitting
>
> A careful look at the numbers in Table 1 in LaRoSA paper [1] already shows that existing papers on activation sparsity also exhibit variance in the performance retained on different tasks for fixed activation sparsity levels. Converting the reported results for CATS [2], TEAL [3], and LaRoSA into percentage deltas reveals that **these methods exhibit the same spread in behavior between tasks** such as Acc-7 and MMLU, even at identical sparsity levels.
>
> | **Method** | **LLaMA2-7B** | | **LLaMA2-70B** | | **LLaMA3-8B** | | **LLaMA3-70B** | | **Qwen2 .5-7B** | | **Qwen2 .5-72B** | | **Mistral-7B** | |
> |:-|:-|:-|:-|:-|:-|:-|:-|:-|:-|:-|:-|:-|:-|:-|
> | | Acc₇ 0-shot | MMLU 5-shot | Acc₇ 0-shot | MMLU 5-shot | Acc₇ 0-shot | MMLU 5-shot | Acc₇ 0-shot | MMLU 5-shot | Acc₇ 0-shot | MMLU 5-shot | Acc₇ 0-shot | MMLU 5-shot | Acc₇ 0-shot | MMLU 5-shot |
> | | | | | | | | | | | |
> | **Dense** | 66 .69 | 45 .85 | 73 .66 | 68 .80 | 70 .05 | 65 .26 | 76 .29 | 78 .71 | 70 .34 | 74 .21 | 75 .58 | 86 .08 | 70 .44 | 62 .34 |
> | | | | | | | | | | | |
> | **CATS 25%** | -4.47% | -6.74% | -1.25% | -1.89% | -3.41% | -5.23% | -0.93% | -0.99% | -0.97% | -2.08% | -3.72% | -1.36% | -8.36% | -4.03% |
> | **TEAL 25%** | -1.12% | -2.6% | -0.48% | -1.6% | -0.93% | -2.16% | -4.01% | -4.89% | -0.82% | -1.35% | -0.7% | -0.74% | -0.54% | -1.33% |
> | **LaRoSA 25%** | -0.45% | -0.41% | -0.38% | -0.09% | -0.73% | -0.63% | -1.22% | -0.3% | -0.31% | -0.63% | -0.07% | -0.53% | -0.27% | -0.85% |
> | | | | | | | | | | | |
> | **CATS 40%** | -25.7% | -46.19% | -14.82% | -18.85% | -21.33% | -51.24% | -7.89% | -8.37% | -12.1% | -13.77% | -9.9% | -3.64% | -15.36% | -28.92% |
> | **TEAL 40%** | -2.65% | -5.21% | -1.62% | -2.94% | -2.73% | -8.31% | -5.31% | -6.96% | -2.32% | -3.73% | -1.23% | -2.32% | -2.39% | -3.48% |
> | **LaRoSA 40%** | -0.81% | -2.6% | -0.48% | -0.93% | -1.8% | -4.06% | -1.15% | -1.38% | -0.95% | -2.53% | -0.3% | -0.64% | -1.42% | -1.91% |
> | | | | | | | | | | | |
> | **TEAL 50%** | -5.2% | -13.7% | -2.36% | -6.35% | -7.32% | -19.12% | -7.03% | -6.31% | -2.86% | -3.73% | -2.32% | -2.61% | -5.27% | -8.02% |
> | **LaRoSA 50%** | -3.12% | -6.0% | -1.09% | -1.79% | -4.08% | -10.13% | -3.25% | -2.8% | -1.78% | -5.55% | -0.53% | -2.02% | -2.81% | -5.68% |
>
> ### Conclusion
>
> We appreciate the Reviewer’s feedback and hope our detailed responses address all the raised points. **The discussed concerns largely reflect inherent limitations of activation sparsity rather than shortcomings of our methodology, and we are confident that our study offers a thorough, practical, and novel analysis within these constraints.** We hope our responses further reinforce the Reviewer's positive assessment of our work and that the Reviewer might consider raising the rating of our paper.
>
> ### References
>
> * [1] *La RoSA: Enhancing LLM Efficiency via Layerwise Rotated Sparse Activation.*, Liu et al., ICML 2025.
>
> * [2] *CATS: Context-Aware Thresholding for Sparsity in Large Language Models*, Lee et al., COLM 2024.
>
> * [3] *Training-Free Activation Sparsity in Large Language Models.*, Liu et al., ICLR 2025.
>
> * [4] Hugging Face documentation for Gemma 3: https://huggingface.co/docs/transformers/main/model_doc/gemma3n
>
> * [5] *Diff-eRank: A Novel Rank-Based Metric for Evaluating Large Language Models.*, Wei et al., NeurIPS 2024.
>
> * [6] *E-rank and the Staircase Phenomenon: New Insights into Neural Network Training Dynamics.*, Yang et al., arxiv 2024

---

> > ### Comment · Reviewer_YvVz · 2025-11-24
> >
> > Thank you for your reply. I believe my questions have been resolved, so I have decided to increase my rating.

---

### Official Review · Reviewer_7CAX · 2025-11-01

**Soundness:** 3
**Presentation:** 3
**Contribution:** 3
**Rating:** 6
**Confidence:** 4

**Summary:**

This paper investigates activation sparsity in modern large models. Beyond ReLU-based networks that produce exact zeros, using SiLU/GELU activations in FFNs yields functional or approximate sparsity, with many activations near zero. The authors address the current fragmented understanding of this phenomenon by introducing a general, training-free framework to evaluate sparsity robustness.

**Strengths:**

They propose a simple top-p sparsification to induce sparsity in various activations (input, gate, up-projection, intermediate). This allows for the introduction of critical sparsity, i.e., the maximum sparsity level that causes less than 1% performance drop. Through extensive experiments on models like Gemma, Llama, and Qwen across different scales (e.g., 1B to 32B parameters), they find that the potential for effective activation sparsity increases with model size. They also find that input-based sparsification is as effective as, or even better than, the more commonly studied gate-based methods, making it a more practical, predictor-free approach. The study also shows that critical sparsity is task-dependent, varying significantly across different downstream evaluations. It also persists across different model types, including instruction-tuned and reasoning-specialized variants. All these provide a comprehensive perspective and practical guidance for harnessing activation sparsity in model design and acceleration.

**Weaknesses:**

This paper's methodology introduces sparsity via the top-p rule and defines critical sparsity as the level at which performance degradation is less than 1%. This is similar to the paper "Sparsing Law: Towards Large Language Models with Greater Activation Sparsity". Further comparison of different sparsity definitions is crucial, as a better definition of sparsity will result in less reduction in model performance.

**Questions:**

Please refer to "Weaknesses".

---

> ### Author Response · Authors · 2025-11-21
>
> We appreciate the Reviewer’s time, constructive feedback, and recognition of our contribution to closing the gap in the existing literature on activation sparsity. Detailed responses to the comments follow below.
>
> > This paper's methodology introduces sparsity via the top-p rule and defines critical sparsity as the level at which performance degradation is less than 1%. This is similar to the paper "Sparsing Law: Towards Large Language Models with Greater Activation Sparsity".
>
> While our approach might seem similar to the one proposed in the mentioned paper [1], there are several important differences between the two.
>
> 1. Our sparsification approach is intentionally designed to be as simple as possible and independent of the data. In contrast, the approach proposed in [1] employs binary search over held-out data to determine the activation threshold per layer under a fixed perplexity constraint, so **the efficiency of their approach depends on the choice of the validation dataset**. Our approach is designed to avoid such dependencies.
>
> 2. The authors of [1] implicitly assume that maintaining the perplexity on language modeling translates to the performance on the downstream tasks, while we sparsify and evaluate performance on the downstream tasks directly. Therefore, **our approach better reflects the task-specific model capabilities for sparsity**. Notably, in [1], Appendix H, the authors admit how their approach fails on more complex tasks. In contrast, our approach obtains high critical sparsity around 80% on zero-shot HumanEval with LLaDA, demonstrating its practical utility for studying activation sparsity on more complex tasks.
>
> 3. **Our study provides broader results for all activation vectors within the FFN module**, while [1] focuses only on the activation sparsity of the intermediate neurons. As we argue in our paper, intermediate sparsity alone offers limited efficiency gains, since it still requires computing roughly two-thirds of the FFN operations.
>
> 4. Finally, **our work provides a more comprehensive and practically relevant view of activation sparsity**, since we evaluate the sparsity properties across a much broader set of models and reach up to 32B parameters, compared to the 1.2B maximum in [1].
>
> > Further comparison of different sparsity definitions is crucial, as a better definition of sparsity will result in less reduction in model performance.
>
> We are not sure if we understand the reviewer correctly here. In our work, we use the term “sparsification rule” to refer to the algorithm used to determine the inactive neuron set, which is required to sparsify activations without exact zero values. Activation sparsity is a ratio of inactive neurons to all possible neurons, and we do not see meaningful alternatives to this definition. We are not familiar with additional sparsity metrics, but we would be open to adding them if the reviewer provides us with references.
>
> As denoted by the Reviewer, there are multiple potential choices for the sparsification strategy, and the choice of this strategy affects the resulting functional sparsity. Therefore, **in Appendix A1, we have already investigated several simple data-free sparsification strategies** we considered at the start of our work, such as top-p, top-k, and max-p. After this initial ablation, we have decided to use top-p because it consistently performed best and remained easy to interpret.
>
> Following the reviewer’s suggestion, **we extended our ablations to a generalized top-p rule**, where we raise activations to some power before applying the top-p operator to compute the sparse mask. We present the additional results obtained with this approach and the motivation behind it in the updated paper, Appendix A2. While there is a small variance in the sparsity characteristics obtained for different powers, **the overall behavior is similar across the operators and further supports our choice of a basic version of top-p** as the primary approach.
>
> ### Conclusion
>
> We hope our answers clarify the unique contributions of our work and address the Reviewer’s concerns. We would be happy to provide additional clarification or discussion if needed.
>
> ### References
>
> * [1] *Sparsing Law: Towards Large Language Models with Greater Activation Sparsity.*, Luo et al., ICML 2025.

---

### Author Response · Authors · 2025-11-21
**Joint response and rebuttal summary**

We thank all Reviewers and the Area Chairs for their time, hard work, and inspiring suggestions. We are glad to note that all the Reviewers have acknowledged the soundness of our approach and praised our comprehensive empirical results. We have responded to all the Reviewers, carefully addressed the raised points, and updated a revised version of the paper, which incorporates the proposed improvements (marked in color). The feedback has significantly helped us improve the paper’s readability, clarity, and practical value.

We summarize the most important rebuttal points and updates made to the paper below:

1. We have extended the paper with an additional analysis of activation sparsity in MoE models in Section 4.5, using Qwen3-30B-A3B as our case study. We have also added Appendix F with additional results for smaller OLMoE-7B-A1B.

2. We significantly improved the readability of the end of Section 4.6 by expanding the analysis of activation sparsity throughout the diffusion process. The added results report sparsity at each diffusion step, providing a clearer context and strengthening the motivation for this experiment.

3. We expanded the analysis of alternative sparsification rules with additional Appendix A2, which includes ablations of the generalized top-p rule.

4. We added Appendix E, which contains a detailed explanation of effective rank computation and makes the paper more self-contained.

5. We clarified the contributions of our work, positioned it more clearly relative to prior research, and addressed several smaller issues raised by the Reviewers, including clarifications on normalization of metrics and the use of the “functional sparsity” term.

We thank all Reviewers once again and remain eager to continue the discussion if needed. We hope our responses address the Reviewers’ concerns and further reinforce the largely positive initial assessment of our work, underscoring the value and relevance of our work.

---

### Author Response · Authors · 2025-12-01
**Discussion summary for the new AC**

We had a productive discussion phase, and the improvements made to the paper led two reviewers to raise their scores. Given that the discussion, to the best of our knowledge, was not influenced by the OpenReview leak, we are very disappointed to see both the rebuttal responses and the score changes rolled back.

Despite this setback, we have continued refining the paper and have updated the paper further to include more comprehensive MoE experiments with an additional OLMoE model. We have also added additional results for MBPP to Table 1, further expanding the scope of our diffusion experiments.

We hope the new AC considers the original, largely positive assessments of our work, together with the substantial improvements made during the discussion phase, and ultimately recommends the acceptance of our paper.

---

### Meta-Review · Area_Chair_qbfG · 2026-01-08

**Summary:**

Although these are purely empirical results and somewhat incremental and even potentially obvious given previous work, the results were sufficiently interesting to several reviewers that it is worth making explicit through experiments. The findings have practical implications and will allow future researchers to harness sparsity more effectively.

**Reviewer Concerns:**

Addressed:
- Analysis of MoE models.
- Comparison to other sparsification strategies.
- PkEJ: Considers conclusion obvious that scale leads to higher critical sparsity. Authors rightly point out that the other reviewers founded an interesting or surprising result, even if it could be predicted from current theory.
- Discussion of differences from Luo et al., ICML 2025.

Outstanding:
- The paper only provides empirical results, and no underlying theory.
- Unclear takeaways from the experiments studying specification across diffusion steps.

**Reviewer Scores:**

The 4 was apparently increased to 6. I doubt the others would have increased. 6/6/6/6

---

### Decision · Program_Chairs · 2026-01-26

Accept (Poster)